# Profiling host ANP32A splicing landscapes to predict influenza A virus polymerase adaptation

Patricia Domingues [1,5], Davide Eletto[1,5], Carsten Magnus [1,2,3,5], Hannah L. Turkington[1], Stefan Schmutz [1], Osvaldo Zagordi[1], Matthias Lenk[4], Martin Beer[4], Silke Stertz[1] & Benjamin G. Hale [1]

Species' differences in cellular factors limit avian influenza A virus (IAV) zoonoses and human pandemics. The IAV polymerase, vPol, harbors evolutionary sites to overcome restriction and determines virulence. Here, we establish host ANP32A as a critical driver of selection, and identify host-specific ANP32A splicing landscapes that predict viral evolution. We find that avian species differentially express three ANP32A isoforms diverging in a vPol-promoting insert. ANP32As with shorter inserts interact poorly with vPol, are compromised in supporting avian-like IAV replication, and drive selection of mammalian-adaptive vPol sequences with distinct kinetics. By integrating selection data with multi-species ANP32A splice variant profiling, we develop a mathematical model to predict avian species potentially driving (swallow, magpie) or maintaining (goose, swan) mammalian-adaptive vPol signatures. Supporting these predictions, surveillance data confirm enrichment of several mammalian-adaptive vPol substitutions in magpie IAVs. Profiling host ANP32A splicing could enhance surveillance and eradication efforts against IAVs with pandemic potential.

[1] Institute of Medical Virology, University of Zurich, Winterthurerstrasse 190, 8057 Zurich, Switzerland. [2] Department of Biosystems Science and Engineering, ETH Zurich, Mattenstrasse 26, 4058 Basel, Switzerland. [3] Swiss Institute of Bioinformatics, Lausanne 1015, Switzerland. [4] Institute of Diagnostic Virology, Friedrich-Loeffler-Institute, Federal Research Institute for Animal Health, 17493 Greifswald-Insel Riems, Germany. [5] These authors contributed equally: Patricia Domingues, Davide Eletto, Carsten Magnus. Correspondence and requests for materials should be addressed to B.G.H. (email: hale.ben@virology.uzh.ch)

nfluenza A viruses (IAVs) infect a broad spectrum of mammalian and avian species, with wild waterfowl and shorebirds being the major natural reservoirs[1]. In recent years, incursions by zoonotic IAV strains, including highly pathogenic avian influenza viruses (HPAIV) of the H5N1 subtype and low-pathogenic H7N9 strains, have continually threatened to cause potentially devastating new human pandemics[2]. However, despite these cross-species transmission events, viral establishment in the new human host is severely limited by a series of powerful host-range restriction barriers to which novel IAVs must first adapt[3]. The IAV RNA-dependent RNA polymerase complex (vPol) is essential for viral genome replication and transcription[4], and is a major site for evolution of host-adaptive viral changes to overcome some of these restrictions[5–10]. ANP32A and ANP32B, two host chromatin regulators, have recently been described as co-factors for vPol function, and a 33 amino-acid insert difference between mammalian (ma) and avian (av) ANP32A orthologs may promote vPol adaptation[11–14]. Indeed, the current model suggests that avian IAV vPols cannot function efficiently with the shorter maANP32A protein (which lacks a hydrophobic sequence related to SUMO-interaction motifs[15]), without prior acquisition of at least one of several key substitutions, such as the typical example of lysine (K) for glutamic acid (E) at position 627 of the vPol subunit, PB2 (E627K)[12]. Adaptation of vPol then leads to increased viral replication, transmission, and virulence in the new host[5,6,16–18]. Nevertheless, some features of this critical and dynamic virus–host interplay remain unresolved. For example, direct experimental evidence is lacking that ANP32A species' differences are the driving force behind selection of disparate residues at PB2-627 or other vPol adaptation sites[19], and host factors such as RIG-I, importins, TUFM, or DDX17 have alternatively been implicated[20–23]. Furthermore, the simple avian vs mammalian ANP32A model does not provide a satisfactory explanation for sporadic observations that IAVs possessing mammalian-adaptive vPol substitutions (including PB2-627K) can be detected in some avian species, such as sparrows or geese, and may be maintained for long periods[24–26].

In this study, we establish experimental tools to dissect whether ANP32A sequence differences alone can drive IAV vPol adaptation. In addition, we apply such systems to understand potential variations in ANP32A across avian and non-avian species, with the aim of identifying specific host conditions that may favor or restrict IAV adaptation.

## Results

**ANP32A splice variants differentially impact vPol activity.** The 33 amino-acid insert in avANP32A after position 175 is a critical and transferable determinant of avian IAV vPol activity[12]. While analyzing non-mammalian ANP32A sequences available in GenBank, we noted that some avian species are predicted to express multiple ANP32A splice variants that differ in composition of the critical insert. This is exemplified by *Gallus gallus* (chickens; Fig. 1a), which possesses a long isoform containing the described 33 amino-acid insert (ANP32A_X1), a shorter isoform with only a 29 amino-acid insert (ANP32A_X2, lacking 4 residues from the hydrophobic SIM-like sequence[15]), and a mammalian-like isoform completely lacking any insert (ANP32A_X3). Using Next-Generation Sequencing (NGS) of an ANP32A cDNA amplicon generated by RT-PCR from the DF-1 chicken fibroblast cell-line[27,28], we confirmed expression of all three splice variants, and determined that ANP32A_X1 is the major variant expressed in these cells (66%), with an abundance at least twice as great as ANP32A_X2 (25%), and 7-times as great as ANP32A_X3 (9%) (Fig. 1a).

To assess whether ability to support avian vPol activity differs between these ANP32A splice variants, we obtained expression constructs representing each ANP32A isoform and used transfection-based polymerase reconstitution assays to compare their abilities to promote model avian-signature (PB2-627E) vPol function. All isoforms could be expressed to similar levels following transient transfection into human 293T cells (Fig. 1b). However, ANP32A_X2 was compromised in its ability to promote PB2-627E vPol function, exhibiting ~50-times less activity than ANP32A_X1 (Fig. 1c, d). ANP32A_X3 (which completely lacks the 33 amino-acid insert) is unable to support avian-signature vPol activity, and behaves like huANP32A[12]. Mechanistically, the defect in ANP32A_X2 function correlated with its reduced capacity to interact with the trimeric avian-signature IAV polymerase complex (Fig. 1e), precisely mimicking the activity and binding phenotypes of an artificial ANP32A_X1 construct with loss-of-function mutations in the SIM-like sequence[15]. Consistent with prior work, only the fully binding-competent ANP32A_X1 could subtly increase expression/stabilization of vPol components (Fig. 1e)[15]. These data support previous observations that the hydrophobic SIM-like sequence in ANP32A_X1 is important for promoting efficient avian-signature IAV vPol activity, likely by enhancing interactions with the viral enzyme[15]. Furthermore, the data reveal that different ANP32A isoforms expressed by avian cells can have disparate effects on avian vPol function.

**ANP32A variants differentially impact IAV replication.** To compare how ANP32A_X1, ANP32A_X2, and ANP32A_X3 support full replication of an avian-signature IAV, we first established a clonal human A549-derived cell-line completely lacking endogenous functional ANP32A expression (A549-ANP32A$_{KO}$) by targeted CRISPR/Cas9 genome editing (Fig. 2a, b). Levels of ANP32B, the closest ANP32A paralog in humans, were not affected by the absence of ANP32A (Fig. 2b). Using lentiviral transduction methods, we stably reconstituted these cells with ANP32A_X1 (ch), ANP32A_X2 (ch), or ANP32A_X3 (hu) (Fig. 2c), and subsequently assessed the propagation abilities of genetically engineered isogenic rWSN-based model IAVs expressing typical avian-signature PB2-627E or typical mammalian-signature PB2-627K. As expected in multicycle growth assays, the PB2-627E virus (but not the PB2-627K virus) was severely attenuated in cells expressing mammalian-like ANP32A_X3. However, the PB2-627E virus replicated similarly to the PB2-627K virus in cells expressing ANP32A_X1. Notably, the PB2-627E virus exhibited subtle attenuation in cells expressing ANP32A_X2 as compared to the PB2-627K virus (Fig. 2d–f). The PB2-627E virus, unlike the PB2-627K virus, was also severely attenuated in the parental A549-ANP32A$_{KO}$ cell-line, confirming the inability of PB2-627E to utilize human ANP32B for replication (Supplementary Fig. 1A)[12]. These observations reveal that ANP32A_X1 and ANP32A_X2, but not ANP32A_X3 (mammalian-like), can alleviate attenuation of model avian-signature vPol expressing viruses, although ANP32A_X2 is not as efficient as ANP32A_X1, particularly at early times post-infection. This broadly correlates with results from our transfection-based assays, where ANP32A_X2 (lacking the hydrophobic SIM-like sequence) exhibited a diminished ability to promote avian-signature vPol activity and to interact with the IAV polymerase complex.

**ANP32A variants differentially select mammal or avian PB2s.** Given the varying abilities of each ANP32A isoform to support replication of the model PB2-627E virus, we sought to assess whether the isoforms consequently differed in their potential to drive mammalian adaptation of an avian-signature IAV. To this

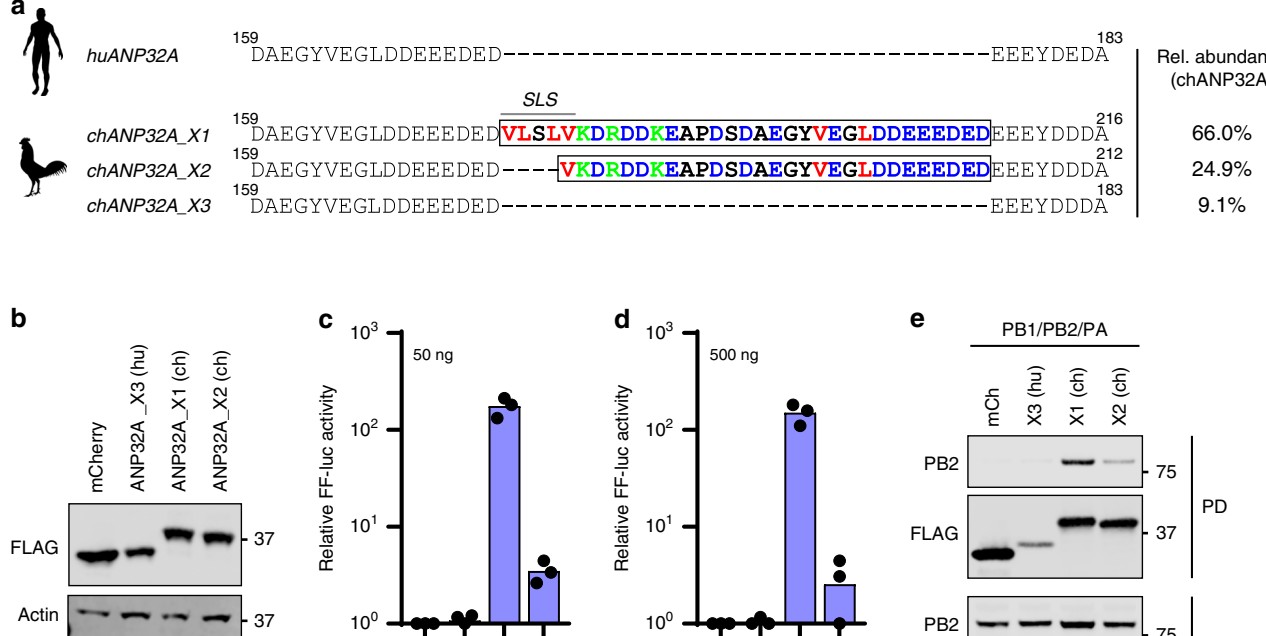

**Fig. 1** Avian species express at least three ANP32A variants that differentially impact avian-signature IAV polymerase activity. **a** Sequence alignment between human ANP32A (huANP32A, *Homo sapiens*, NP_006296.1) and chicken ANP32A (chANP32A, *Gallus gallus*_X1, XP_413932.3; *Gallus gallus*_X2, XP_004943985.1; *Gallus gallus*_X3, XP_025009881.1). Hydrophobic residues (red), acidic stretches (blue) and basic residues (green), are highlighted. SLS, SIM-like sequence. Relative abundance of ANP32A isoform transcripts in chicken DF-1 cells, as determined experimentally by NGS quantification of cDNA-derived amplicons, is shown on the right. **b** Western blot analysis of lysates from human 293T cells transfected with the indicated FLAG-ANP32A constructs. **c** Polymerase reconstitution assay comparing the impact of each FLAG-ANP32A construct (50 ng) on PB2-627E vPol activity in human 293T cells. **d** Similar to **c**, but using 500 ng of each FLAG-ANP32A construct. In panels **c**, **d**, bars represent mean values from three independent experiments, with the individual data points shown. **e** 293T cells were transfected with the indicated FLAG-tagged constructs together with PB1, PA, and PB2 (627E). Following anti-FLAG precipitation (PD), the indicated proteins were detected by western blot. For panels **b** and **e**, representative data from two independent experiments are shown. Source data for panels **b**–**e** are provided in the Source Data file

end, we established a competition-based NGS assay involving low multiplicity-of-infection (MOI) serial passaging of a defined mixture of rWSN-based PB2-627K and PB2-627E model viruses to determine which PB2 variant would be selected for by each ANP32A isoform. To simulate the 'real-life' scenario found in most avian species, where viruses expressing avian adaptations such as PB2-627E are dominant prior to mammalian adaptation, we biased the system against PB2-627K by using an E:K ratio of 5:1. Passaging revealed that ANP32A_X1 expression could enrich for the PB2-627E population (Fig. 2g), identifying a general fitness cost to the PB2-627K virus in this 'avian-like' cell model that was not detectable by the less-sensitive multicycle growth analysis titrated by plaque assay (Fig. 2d), and mimicking the situation in most birds. This effect was clearly driven by the overexpression of ANP32A_X1 in this system, as PB2-627K was selected for in the parental A549-ANP32A$_{KO}$ cell-line, indicating that expression of ANP32A variants can titrate out the effect of endogenous ANP32B (Supplementary Fig. 1B). Strikingly, despite different kinetics, both ANP32A_X2 and ANP32A_X3 selected for PB2-627K in this system (Fig. 2h, i). Similar results were obtained when competition assays were performed with virus mixtures where the input E:K ratio was 99:1 (Supplementary Fig. 1C–E), albeit this extremely low input amount of PB2-627K permitted slow de novo selection of alternative mammalian-adaptive mutations in PB2, such as 630R[29] (Supplementary Fig. 1F, G). This confirms that ANP32A_X2 and ANP32A_X3 must be

partially or wholly defective, respectively, at supporting avian-signature vPol activity, and therefore have the capacity to drive efficient selection of mammalian-adaptive vPol substitutions. These data provide direct experimental evidence to support previous findings that ANP32A alone is a key determinant of IAV vPol host restriction[12], and uncover that different ANP32A splice variants can drive selection of critical species' adaptations in the IAV PB2 gene.

**ANP32A splicing ratios vary across avian species.** Given our data that different ANP32A splice variants can functionally impact host adaptation of IAVs, we surveyed the ANP32A splicing landscapes of various species. We aimed to understand if ANP32A splicing differences exist between potential avian IAV hosts that could explain sporadic observations of mammalian-adaptive vPol substitutions in viruses isolated from certain bird species, including tree sparrows, pigeons, geese, and swans[24–26]. Such an explanation would be an extension of the finding that mammalian-adaptive PB2 substitutions can occur in ratites, such as ostriches, emus and rheas, which only have an X3-like ANP32A variant[12,30]. To this end, we generated ANP32A cDNA amplicons from mRNA extracted from cells of different mammalian and avian species, and used NGS to estimate the abundance of each X1-like, X2-like, X3-like, and other novel ANP32A spliced isoforms (corresponding to the chicken orthologs; see

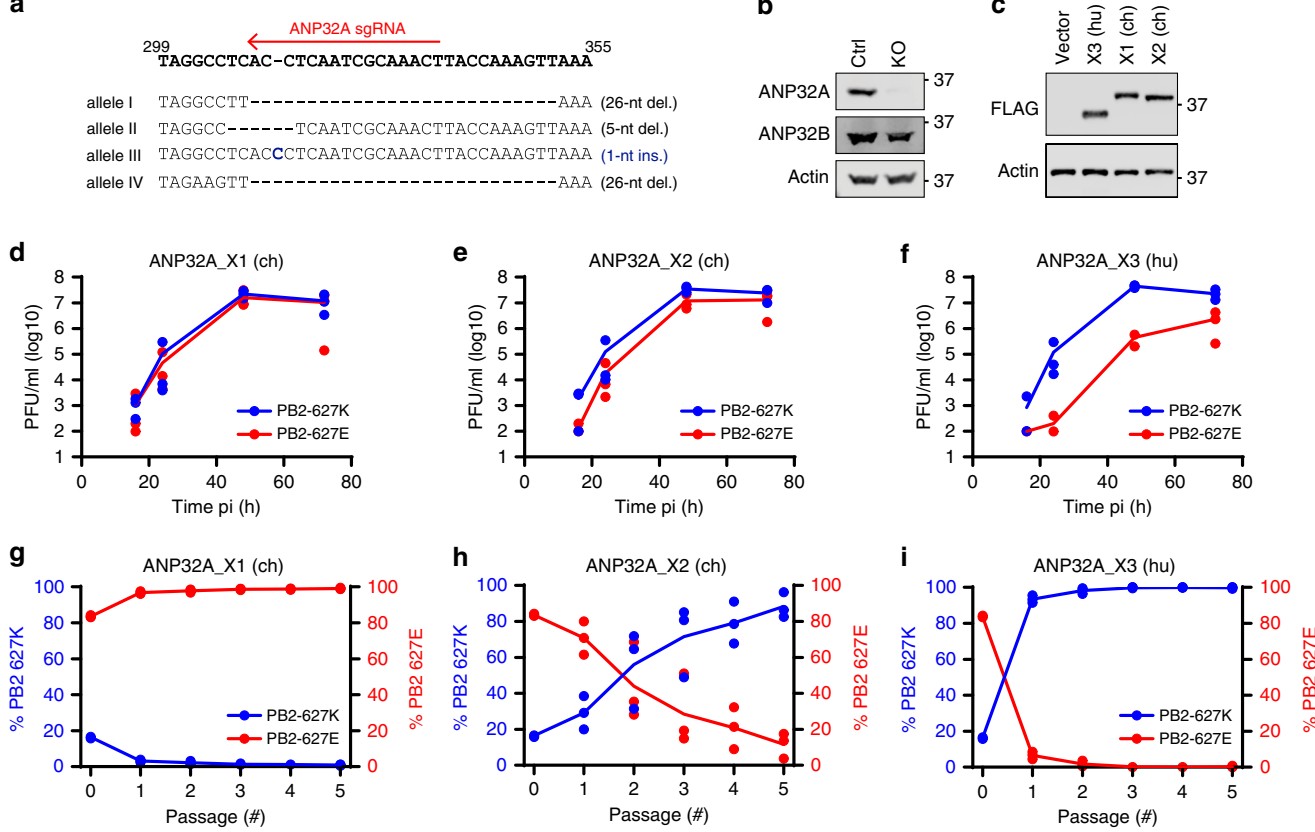

**Fig. 2** ANP32A variants differentially impact avian-signature IAV replication and selection. **a** Genotype of the A549-ANP32A_KO cell-clone used in this study. The target site of the crRNA is depicted. Four distinct ANP32A alleles were detected in A549s suggesting duplication of chromosome 15. **b** Western blot analysis of parental A549 (Ctrl) and A549-ANP32A_KO (KO) cells for ANP32A, ANP32B, and actin. **c** Western blot analysis of A549-ANP32A_KO cells stably reconstituted with the indicated FLAG-ANP32A constructs or empty vector. **d**–**f** Viral growth kinetics of rWSN-based viruses expressing PB2-627E or PB2-627K in the indicated ANP32A-reconstituted cell-lines (MOI = 0.001 PFU/cell). **g**–**i** Competition assays between rWSN-based viruses expressing PB2-627E or PB2-627K (5:1 input ratio) in the indicated ANP32A-reconstituted cell-lines. P0 represents input. Percentage of PB2-627K/E was determined by NGS. In panels **d**–**i**, mean values from three independent experiments are plotted with lines, and the individual data points are shown. For panels **b** and **c**, representative data from two independent experiments are shown. Source data for panels **b**–**i** are provided in the Source Data file

Supplementary Table 1). To promote accuracy, we ensured that our sequenced amplicons had a total read depth greater than 50,000 reads, and only considered those splice variants with a coverage of at least 0.5% of the total read count. As expected, all mammalian species tested expressed only a single ANP32A_X3-like isoform. Horse was an exception, which expressed an additional low-abundance X3-like isoform that lacks a portion of the Low Complexity Acidic Region (LCAR; Fig. 3a). In contrast, gulls (a major reservoir species for many avian IAV subtypes) exclusively expressed a single ANP32A_X1-like isoform, while all other avian cells expressed multiple ANP32A isoforms that varied greatly in relative abundance between species. For example, in chicken, duck, and turkey, ANP32A_X1 was clearly the most abundant isoform (62–83%), followed by ANP32A_X2 (15–29%) and ANP32A_X3 (2–11%). A similar relative abundance was found in quail, except ANP32A_X3 was not detected in this species. However, while ANP32A_X1 was also the most abundant isoform in the four cell-lines derived from goose and swan, it was striking that in these species ANP32A_X3 (24–28%) dominated over ANP32A_X2 (6–9%; Fig. 3a, Supplementary Table 1). Notably, the passerine birds (swallow, blackbird, and magpie) were the only avian species examined where ANP32A_X1 was not the most abundant isoform expressed: ANP32A_X3 expression dominated in swallows; ANP32A_X2 dominated in blackbirds; and all three isoforms were equally abundant in magpies (Fig. 3a, Supplementary Table 1). This variability in ANP32A

splice variant abundance does not appear to be related to tissue-specificity or transformation status of cells, as the isoform ratio was stable across different chicken tissues derived from healthy adult animals and from egg chorioallantoic membranes (Fig. 3a, b, Supplementary Table 2), and did not vary between primary chicken cell cultures and different chicken cell-lines (Fig. 3a). Furthermore, two independent duck cell-lines and three independent goose cell-lines exhibited the same intra-species ANP32A splice variant ratios, supporting a species-specific determinant to ANP32A splicing landscapes (Fig. 3a). Notably, the relative ANP32A splice variant ratios did not change in chicken cells during the stress of IAV infection (Fig. 3c). However, we did find that temperature had a marked effect on the chicken cell ANP32A splicing landscape, with cooler culturing temperatures promoting expression of ANP32A_X2 and ANP32A_X3 at the expense of ANP32A_X1 (Fig. 3d). Phylogenetic analysis of the avian species examined, based on the determined ANP32A_X1 sequences, revealed that splicing landscapes also mirrored evolutionary relationships, with passerine birds grouping separately from the others and land-based poultry (chicken, turkey, and quail) grouping together (Fig. 3e). Overall, these data indicate that birds express a wider range of ANP32A isoforms than previously appreciated (including splice variants suboptimal for avian-signature IAV vPol activity), and the expression ratios of these isoforms can vary greatly between different avian species.

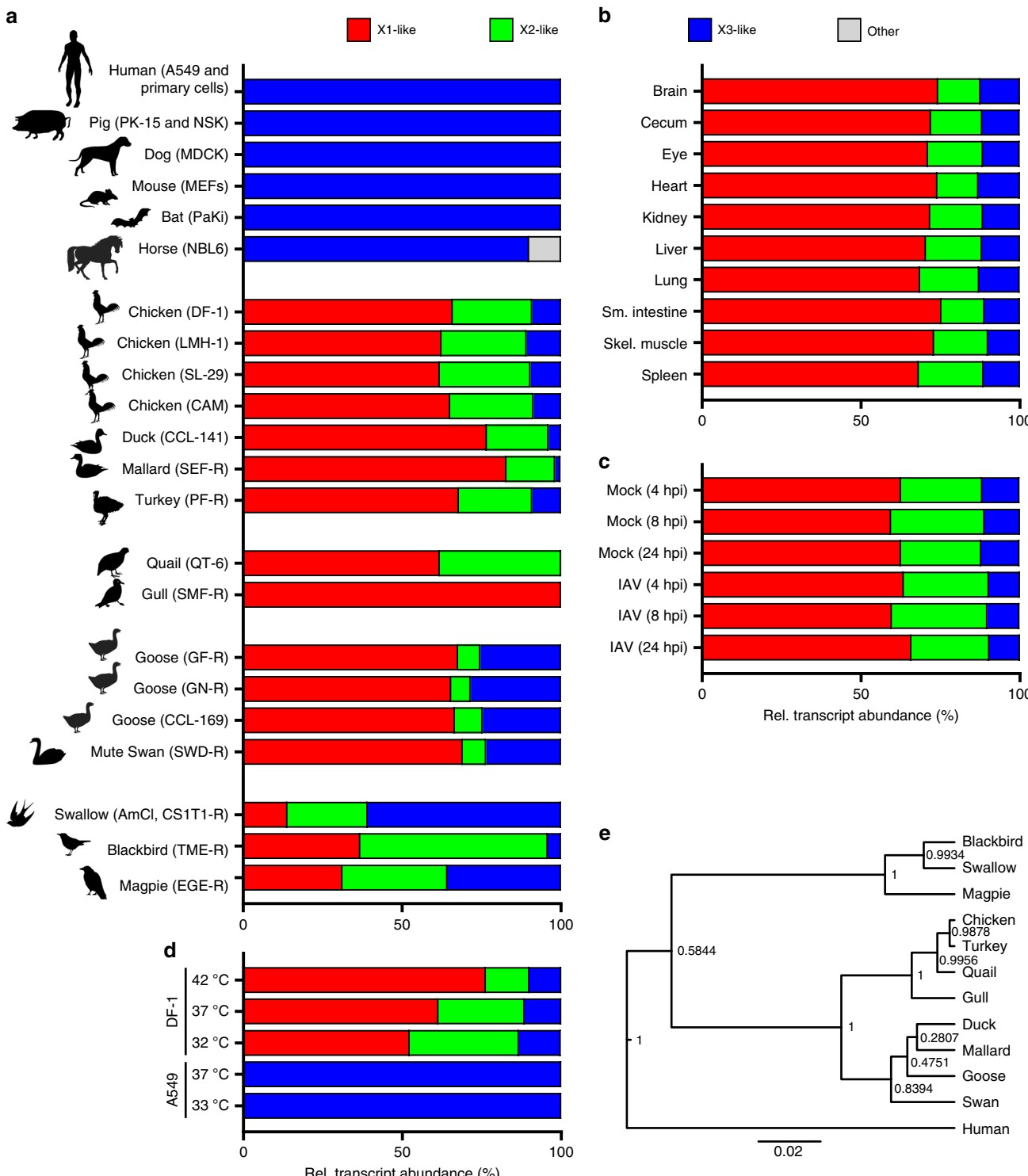

**Fig. 3** ANP32A splicing ratios vary across species, and can be affected by temperature. **a** Relative abundance of ANP32A X1-, X2-, and X3-like isoform transcripts in cells from various mammalian and avian species, as determined experimentally by NGS quantification of cDNA-derived amplicons. **b–d** Relative abundance of ANP32A X1-, X2- and X3-like isoforms in: **b** different tissues derived from a healthy adult chicken; **c** uninfected and infected chicken DF-1 cells (WSN, MOI = 1 PFU/cell); **d** chicken DF-1 and human A549 cells cultured for at least 72 h at the indicated temperatures. In all panels, data are representative of at least three replicate samples. See also Supplementary Tables 1 and 2. **e** Phylogenetic analysis of the determined ANP32A_X1 sequences from each species to infer evolutionary relationships. The plot shows the maximum clade credibility tree with posterior confidence of splitting events at the nodes of the tree. The scale bar represents the number of nucleotide substitutions per site

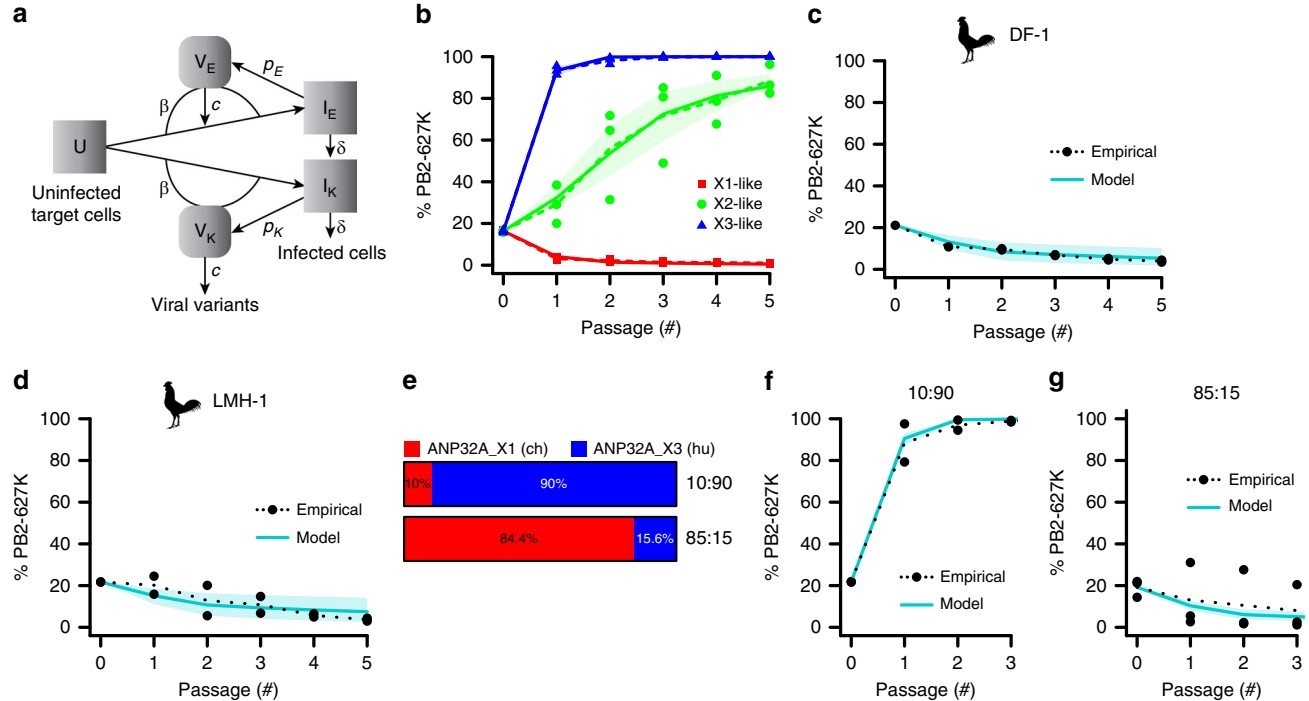

**Fig. 4** Modeling and validating the impact of different ANP32A splice variant ratios on selection of mammalian- or avian-like IAV adaptations. **a** Diagrammatic representation of the mathematical model used in this study. **b** Modeling the competition-based passaging assays between isogenic viruses expressing PB2-627E or PB2-627K in the indicated ANP32A-reconstituted A549 cell-lines (dotted colored lines: original data average, with original data points; colored solid lines: model of best estimates of virus production rates, with lighter areas indicating the 95% bootstrap confidence intervals of these estimates). **c, d** Model prediction and empirical passaging results in chicken DF-1 (**c**) or LMH-1 (**d**) cell-lines. The average of two independent replicates is indicated with a dotted line (original data points shown), the model prediction with a solid line, and the 95% bootstrap confidence interval of the model with a shaded area. **e** NGS analyses of A549-ANP32A$_{KO}$ cells engineered to express different ratios of FLAG-tagged ANP32A_X1 and ANP32A_X3. **f, g** Model prediction and empirical passaging results in the engineered cells described in **e** (lines analogous to the scheme used in **c**, **d**; **f** two independent replicates; **g** three independent replicates). Source data for panels **c**, **d**, **f** and **g** are provided in the Source Data file

**Modeling the impact of ANP32A splicing on IAV adaptation.** We developed a mathematical model to predict the effect of different ANP32A splice variant ratios (as found in different avian species) on selection of mammalian vPol adaptations during IAV replication. This model is an extension of one previously described for general IAV dynamics[31], but specifically accounts for two viral variants that differ only in their vPol use of ANP32A (e.g. PB2-627K/E), and which therefore infect target cells at identical rates. However, subject to the relative abundance of each possible ANP32A splice variant, the infected cells are modeled to produce new virus at rates defined as $p_K$ or $p_E$, depending upon the respective input variant (Fig. 4a and Methods). We used our experimental data to estimate virus production parameters for each viral variant with each ANP32A isoform. Our derived model faithfully recapitulated the original observations that ANP32A_X1 can select for avian vPol adaptations (such as PB2-627E), while ANP32A_X2 and ANP32A_X3 select for mammalian vPol adaptations (such as PB2-627K; Fig. 4b). In addition, fitting the mathematical model to the ANP32A_X1, ANP32A_X2, and ANP32A_X3 passaging data led to estimates of the virus production rates of PB2-627K/E viruses in cells expressing solely one of the three ANP32A splice variants. We then sought to establish whether feeding a weighted average of these rate estimates (based on the relative abundance of ANP32A splice variants) into our model could predict PB2-627K/E selection in cells expressing different ratios of ANP32A splice variants (see Methods). We validated this approach by performing competition experiments using the model rWSN-based PB2-627K and PB2-627E viruses in chicken DF-1 and LMH-1 cells, which express endogenous ANP32A X1, X2, and X3 isoforms at ratios of 66:25:9

and 62:27:11, respectively. We found that the empirically derived PB2 variant selection patterns correlated very well with the model predictions for these avian cell-lines based on the weighted average approach (Fig. 4c, d). To further experimentally validate our system, we used double lentiviral transductions to select two independent cell populations (in the A549-ANP32A$_{KO}$ background) reconstituted with defined ratios of ANP32A_X1 (ch) and ANP32A_X3 (hu), namely X1:X3 at 10:90, and X1:X3 at 85:15 (Fig. 4e). Subsequent competition experiments using the model PB2-627K and PB2-627E viruses revealed distinct PB2 variant selection patterns in each engineered cell-line that also correlated well with the selection pattern predicted by the model (Fig. 4f, g). Overall, these data support the hypothesis that ANP32A variant ratios, rather than absolute levels of any particular variant, can drive IAV vPol adaptation in mammalian and avian cells, and that knowledge on ANP32A splicing ratios can be used to broadly predict adaptive routes.

**Using ANP32A splicing to predict adaptation.** We applied our experimentally validated algorithm to all species for which we obtained ANP32A splice variant expression data, modeling the expected impact that each host might have on selection of IAVs bearing mammalian adaptations in vPol. From these models, it was clear that all mammalian and ratite species (expressing only ANP32A_X3) are predicted to provide a high selection pressure favoring rapid acquisition of mammalian-like vPol adaptations (Fig. 5a), which has previously been observed empirically[30]. In contrast, and as expected, most avian species are predicted to favor selection of avian-like vPol adaptations (Fig. 5b–g,

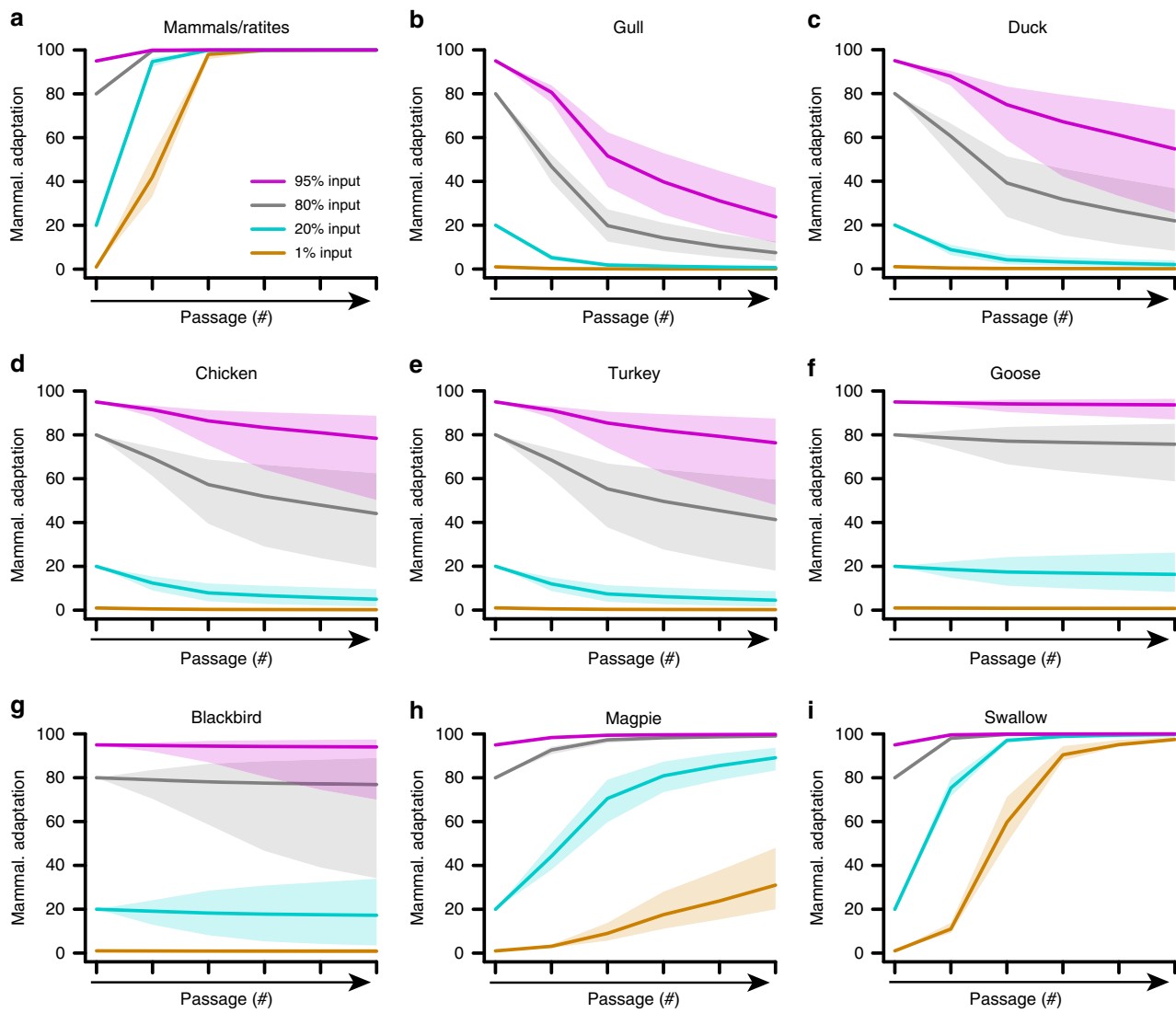

**Fig. 5** Modeling the impact of species' ANP32A splice variant ratios on selection of mammalian- or avian-like IAV adaptations. **a–i** Model predictions of mammalian-like vPol selection in the indicated species, taking into account their ANP32A splice variant ratios as shown in Fig. 3a. Each input mammalian-like vPol percentage assessed is indicated by a separate color. The solid lines indicate the model prediction with the best estimates of virus production rates, and the lighter areas show the 95% bootstrap confidence intervals

Supplementary Fig. 2). Strikingly, swallows and magpies were clear exceptions: both these species are characterized by a low abundance of ANP32A_X1 relative to ANP32A_X3, therefore the model predicts that they might promote selection of mammalian-like vPol adaptations (Fig. 5h, i). A notable additional observation from our model outputs was that the rates of selection predicted for each avian species varied considerably. For example, gull, duck, chicken, and turkey appeared to be rapid avian-like vPol adaptation selectors, while species such as blackbird, goose, and swan were much slower, and could even be relatively neutral with regard to selection (Fig. 5b–g, Supplementary Fig. 2B).

By calculating the area under the passaging prediction curve (AUC) relative to input for all species for which we obtained ANP32A splice variant ratios, we determined risk scores relating to the potential for a particular species to act as a driving force to select for mammalian-like adaptations in vPol (Fig. 6a). The risk scores summarize the modeling output, and can be used to predict species that may select for classical mammalian-like vPol adaptations (mammals, ratites, swallow, and magpie), or classical avian-like vPol adaptations (blackbird, goose, swan, turkey, quail, chicken, duck, and gull). As the mathematical model for

passaging prediction – and therefore also these risk scores – is dependent on several input parameters and conditions for estimating virus production rates, we performed three different sensitivity analyses: (i) varying conditions for estimating virus production rates (Supplementary Fig. 3A); (ii) varying infection rate ($\beta$), rate of infected cell death ($\delta$), and rate of viral clearance ($c$) (Supplementary Fig. 3B); and (iii) relaxing assumptions that these three rates are the same for each virus variant (Supplementary Fig. 3C). All potential modeling scenarios broadly supported our risk estimates for which species might select for, or maintain, mammalian-like vPol adaptations. In addition, we integrated our model and risk analysis output into a user-friendly web-based platform that allows individual variation of parameter estimates, as well as input of ANP32A splicing ratios (see Methods). This tool can be used to identify threshold values for ANP32A splicing ratios that likely drive mammalian versus avian vPol adaptations (e.g. <30% ANP32A_X1 always selects for mammalian-like adaptations in vPol; Fig. 6b), to vary model parameters, or to input newly determined ANP32A splice variant ratios from different species in the future and predict their propensity for selection.

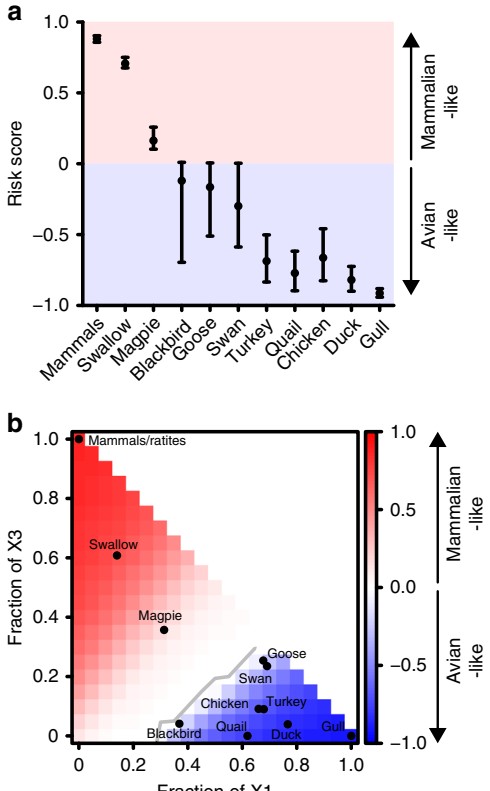

**Fig. 6** Evaluating the risk of different species selecting for mammalian-like IAV adaptations. **a** Estimated relative risk scores for each indicated species to select for avian- or mammalian-like IAV vPol adaptations. The dots indicate the risk score prediction based on the best estimates of virus production rates alongside the 95% bootstrap confidence intervals (error bars). **b** Heat-map of risk scores based on the mathematical model and using the best estimates of virus production rates. Axes indicate fraction of the total ANP32A complement represented by X1 or X3, allowing threshold values for selection (gray line) to be determined

**Enrichment of mammalian adaptations in magpie IAVs.** Poor replication of our model IAV strain in the primary swallow and magpie cells hampered efforts to directly test these two species as experimental selectors of mammalian-like vPol adaptations in vitro. To circumvent this problem, as well as to assess our predictions in a 'real-life' in vivo setting, we took a natural surveillance approach and analyzed over 8000 viral sequences obtained from different mammalian and avian species that are stored in the NIAID Influenza Research Database (IRD)[32]. We focused on H5, H7, and H9 IAVs to limit any bias from sporadic spill-over events of mammalian-adapted viruses into birds, and identified experimentally validated mammalian vPol adaptation sequences[19,33] in IAVs isolated from magpies. The total number of available magpie IAV sequences are relatively low (n = 12) due to a lack of surveillance in this species. We did not analyze sequences from swallows as only four such IAVs have been isolated to date. Nevertheless, we compared the prevalence of these known mammalian-like vPol adaptations in IAVs isolated from humans, swine, ratites (emus, ostriches, and rhea), geese, swans, turkeys, quails, chickens, ducks, and gulls. Using the surveillance results from gulls as a benchmark for 'typical' avian-like vPol adaptation (as wild waterfowl and shorebirds are the major natural reservoirs for IAVs[1]), we found that only H5, H7, and H9 IAVs naturally isolated from humans and ratites exhibited a significant enrichment for the typical mammalian vPol adaptation PB2-627K[30] (Fig. 7a). PB2-701N was only found to be significantly enriched in human isolates (Fig. 7b). However,

other experimentally validated mammalian-like vPol adaptations, such as PB2-391Q, PB2-456D[19], and PA-100I[33] are significantly enriched in magpie IAVs as compared to gull IAVs (Fig. 7c–e). Notably, PB2-456D and PA-100I are also enriched in swine and human IAVs, respectively, providing additional evidence for their likely involvement in mammalian vPol adaptation (Fig. 7d, e). These surveillance data identifying mammalian-adaptive vPol substitutions in magpie IAVs support the validity and applicability of our mathematical model for predicting adaptation routes.

## Discussion

Herein, we show that cells from many different avian species can express at least three ANP32A splice variants (X1, X2, and X3) that differ only in composition of the avian IAV vPol-supporting insert. Notably, the relative expression ratios of these three ANP32A variants are highly species-specific, a finding corroborated by Baker et al.[34] while our manuscript was in preparation. A variant (ANP32A_X2) that lacks four amino-acid residues in the hydrophobic SIM-like sequence is approximately 50-times less efficient than the 'full-length' ANP32A_X1 at promoting avian-signature IAV vPol activity, which correlates with its reduced ability to interact with this complex. Concomitantly, the ability of ANP32A_X2 to support full replication of an avian-signature IAV is also partially compromised. Strikingly, expression of this less active ANP32A_X2, or the inactive ANP32A_X3, restricts efficient avian-signature IAV replication to such an extent that both isoforms are capable of driving selection of IAVs with mammalian-like adaptations in the viral polymerase (i.e. PB2-627K in our experimental system). Using these experimental kinetic data, we modeled (and validated) predictions relating to how different ratios of ANP32A isoforms can impact vPol adaptation. Applying this model to data from disparate avian species, we show how specific ANP32A splicing landscapes could potentiate acquisition, or maintenance, of vPol substitutions that might ultimately 'pre-adapt' IAVs to mammalian cells, a notion also suggested by Baker et al.[34]. Remarkably, passerine species such as swallow and magpie, are predicted to be drivers of mammalian-like adaptations in vPol, and indeed, several experimentally validated markers of mammalian-like vPol adaptation appear to be enriched in the few available magpie IAV sequences already collected in limited surveillance studies. Our findings therefore suggest that it might be prudent to expand surveillance efforts for zoonotic IAVs of mammalian concern into additional avian (passerine) species, and to combine this with a broad survey of inter-species ANP32A splice ratios. Our modeling also predicts that blackbirds, geese, and swans might maintain sporadic mammalian-adaptive vPol mutations for longer periods of time than species such as gulls, ducks, or chickens. These observations are noteworthy as some of these potential wild bird 'maintainer' species are migratory, and therefore could disseminate avian IAVs with critical 'pre-pandemic' markers to other species along their migratory fly-ways, a situation that has been observed previously with bar-headed geese[24]. The identification of new, perhaps previously unappreciated, avian hosts that can precipitate at least some mammalian adaptations could be used to improve risk assessment and management strategies to limit contact of such species with avian IAV sources. This is an important consideration as swallows, magpies, and related passerines have ample opportunities to interact with both poultry and swine on farms, and could act as intermediate hosts promoting the gradual adaptation of an avian IAV to mammals. In sum, our functional description of ANP32A splicing landscapes, and the adaptive forces they exert, may provide a new molecular framework to understand and control new sources that contribute to evolution of mammalian-adaptive mutations in avian IAVs,

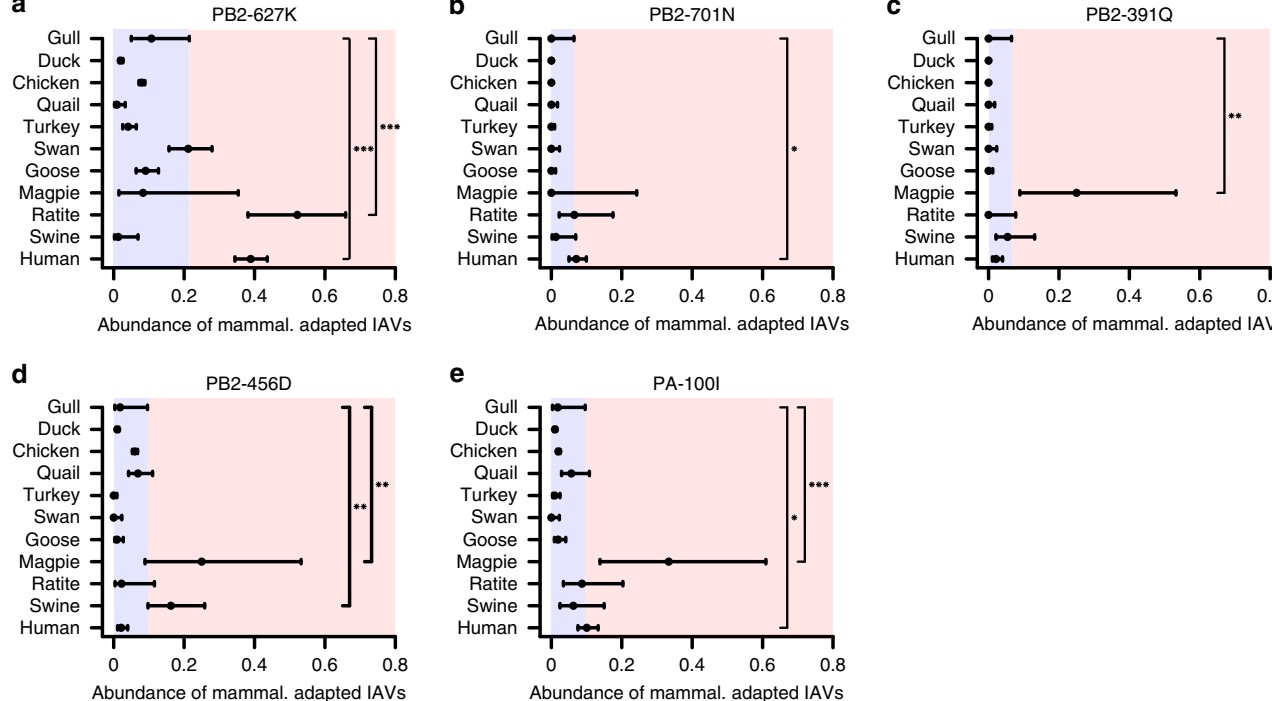

**Fig. 7** Analysis of surveillance data suggests enrichment of mammalian-adaptive vPol substitutions in IAVs isolated from magpies. **a–e** Abundance of mammalian-adaptive vPol sequences in H5, H7, and H9 IAVs isolated from the indicated species: **a** PB2-627K; **b** PB2-701N; **c** PB2-391Q; **d** PB2-456D; and **e** PA-100I. Error bars indicate 95% confidence intervals of the abundance of the specific substitution in the species population. Blue shading broadly highlights the range expected in species expressing only ANP32A_X1, thus indicating substitution abundances typically found in species selecting for avian-like vPol variants, while red shading broadly highlights the substitution abundance expected in species selecting for mammalian-like vPol variants. Stars indicate that the identified mammalian-like vPol substitutions are significantly higher in abundance than those found in species expressing only ANP32A_X1 (i.e. gull). Significance was determined by a one-sided proportion test (*0.01 < p < 0.05; **0.001 < p < 0.01; ***p < 0.001). All other comparisons were non-significant. Source data, including sample sizes, for all panels are provided in the Source Data file

and their consequences for cross-species transmission and human pandemic emergence.

## Methods

**Cells**. All mammalian and avian cells used, together with their source, are described in Supplementary Table 3, and were cultured as detailed by the appropriate source datasheet.

**CRISPR-Cas9 genome editing**. To generate a clonal A549 cell-line lacking expression of endogenous ANP32A (A549-ANP32A$_{KO}$), we used a ribonucleoprotein (RNP)-based system consisting of Alt-R S.p. Cas9 nuclease (IDT, #1074181) in complex with Alt-R CRISPR-Cas9 crRNA (AltR1/rGrU rArArG rUrUrU rGrCrG rArUrU rGrArG rGrUrG rGrUrU rUrUrA rGrArG rCrUrA rUrGrC rU/AltR2, IDT) and tracrRNA (IDT, #1075927). In brief, pre-assembled RNP complexes were delivered into A549s by reverse transfection with RNAiMax (ThermoFisher Scientific), and individual cell clones were generated by limiting dilution. An initial round of phenotype screening was performed by immunoblotting for ANP32A in whole cell extracts. Genotypes of selected clones were then determined by NGS. In brief, gel-extracted, Illumina-compliant amplicons were analyzed by MiSeq (Illumina), with 260 cycles of read 1, 8 cycles of index 1, and 8 cycles of index 2. The following primers were used for generation of the NGS amplicon: NGS-Fw, CTT TCC CTA CAC GAC GCT CTT CCG ATC T ATG AAG GCA AAC TCG AAG GC; NGS-Rv, GAC TGG AGT TCA GAC GTG TGC TCT TCC GAT CT GTC TGG CAT GTT GGT GCA AG (underlined indicates priming sites for 2nd round PCR primers, D50x and D70y TruSeq HT adapters, respectively).

**Plasmids**. The expression plasmids for FLAG-tagged mCherry, huANP32A, chANP32A_X1, and the IAV polymerase components and appropriate reporter constructs have been described[15,35]. All additional gene variants were generated by Quikchange II XL Site-Directed Mutagenesis (Agilent Technologies). Indicated constructs were also sub-cloned into pLVX-IRES-Puro or pLVX-IRES-Neo (Clontech) and used to produce lentiviral stocks as described below. New constructs were authenticated by DNA sequencing. All plasmid transfections were performed using Fugene HD (Promega) at 1:3 DNA:transfection reagent ratio.

**Generation of cells expressing ANP32A variants**. Lentiviral stocks were prepared by co-transfecting 293T cells with each pLVX-IRES-Puro/Neo-based plasmid, together with pMD2.G and pCMVdR8.91. Lentiviral supernatants were harvested 60 h post-transfection, filtered and aliquoted, and stored at −80 °C. To generate polyclonal cell-lines expressing FLAG-ANP32A variants, A549-ANP32A$_{KO}$ cells were transduced with the appropriate lentivirus stock for 48 h in the presence of 8 μg/mL of polybrene (Millipore) prior to selection with puromycin and/or neomycin.

**Polymerase assays, immunoprecipitations, and western blots**. Polymerase reconstitution assays, vPol interaction assays (co-immunoprecipitations), and western blot analyses were all performed using standard described methods[15]. FLAG-tagged constructs were detected using FLAG M2 antibody F1804 (Sigma; 1:2000 dilution), PB2 was detected using a custom rabbit polyclonal anti-serum (1:2000 dilution), PA was detected using antibody GTX118991 (Genetex; 1:2000 dilution), actin was detected using antibody A2103 (Sigma; 1:3000 dilution), and ANP32A and ANP32B were detected using antibodies ab51013 and ab184565 (Abcam; 1:1000 dilution), respectively.

**Construction and analysis of recombinant viruses**. Recombinant A/WSN/33 (H1N1) (rWSN) viruses were rescued and titrated as described[36,37] with minor modifications. In brief, 6 × 10$^5$ 293T cells were seeded in 6-well plates and co-transfected 24 h later with eight ambisense pDZ-based expression plasmids encoding all WSN segments: nucleoprotein (NP), PA, PB1, PB2 (627 K or 627E variants), matrix (M), hemagglutinin (HA), neuraminidase (NA), and NS (gift from Adolfo García-Sastre (Icahn School of Medicine at Mount Sinai, USA)). A plasmid expressing chicken ANP32A was co-transfected together with the appropriate rescue plasmids while generating the PB2-627E virus. Twenty-four hours post-transfection, cells were washed once in sterile phosphate-buffered saline (PBS) and 3 × 10$^5$ MDCK cells (stably-expressing chicken ANP32A for the PB2-627E virus) were added in DMEM supplemented with 1 μg/mL tosylsulfonyl phenylalanyl chloromethyl ketone (TPCK)-treated trypsin (Sigma-Aldrich, MO). Forty-eight hours later, supernatants were harvested, viruses plaque-purified, and virus stocks grown and titrated using standard methods in MDCK cells (stably-expressing chicken ANP32A for the PB2-627E virus). RNA was extracted from stock aliquots using the ReliaPrep$^{TM}$ RNA Tissue Miniprep System (Promega), and the PB2 genomic segments of each virus were fully sequenced

after segment-specific reverse transcription-PCR (RT-PCR) to ensure absence of undesired mutations.

To determine virus replication kinetics, $4 \times 10^5$ of the indicated cells were seeded in 12-well plates and infected 24 h later with each virus diluted in PBS supplemented with 100 units/mL penicillin, 100 μg/mL streptomycin (Gibco Life Technologies), 0.45% bovine albumin (Sigma-Aldrich), and 1 mM $Ca^{2+}/Mg^{2+}$. A multiplicity of infection (MOI) of 0.001 plaque-forming units (PFU)/cell was used. Following 1 h of adsorption, cells were washed three times in PBS before being incubated in DMEM supplemented with 100 units/mL penicillin, 100 μg/mL streptomycin (Gibco Life Technologies), 0.45% bovine albumin (Sigma-Aldrich), 0.1% FBS, and 20 mM HEPES (Gibco Life Technologies). Supernatants were harvested at the indicated time points and titrated by standard plaque assay on MDCK cells.

To perform passaging assays, $4–6 \times 10^5$ of the indicated cells were seeded in 12-well plates and infected 24 h later with a mixed infectious virus inoculum (PB2-627E:K at 5:1 or 99:1 as indicated) at an MOI of 0.001 PFU/cell (A549 based assays) or 0.01 PFU/cell (DF-1 and LMH-1 based assays). Following 1 h of adsorption, cells were treated as above. Supernatants were collected 48 h post-infection (passage 1), and used for repeated infection of new cells at the estimated same MOI to generate passage 2. This procedure was repeated until the indicated passage. RNA from supernatants of each passage, and input inoculum, were extracted using the ReliaPrep™ RNA Tissue Miniprep System (Promega). Viral RNA was subjected to one-step RT-PCR (AccessQuick™ RT-PCR System, Promega) to generate NGS-suitable amplicons centered around the PB2-627 region. Primers used for generation of the NGS amplicon were: PB2-627-NGS-F, CTT TCC CTA CAC GAC GCT CTT CCG ATC TAT GAG GGA TGT GCT TGG GAC; and PB2-627-NGS-R, GAC TGG AGT TCA GAC GTG TGC TCT TCC GAT CTT GCG GAC TCA ACT CCA GCT G (underlined indicates priming sites for 2nd round PCR primers, D50x and D70y TruSeq HT adapters, respectively). NGS outputs were analyzed in two steps: firstly, each read was aligned to the wt WSN reference sequence by Burrows-Wheeler Aligner;[38] secondly, the frequency of each variant of interest (e.g. the codon encoding 627E/K) was determined by the variant caller LoFreq[39], filtering out calls with a coverage depth <500 reads and allele frequency (AF) <0.005.

**Next-generation sequencing of ANP32A.** To determine the relative abundance of ANP32A_X1, ANP32A_X2, and ANP32A_X3 transcripts, cDNA was synthesized from 1 μg of total RNA extracted from each cell-line (treated as indicated), primary chicken chorioallantoic membrane, or healthy chicken tissue (AMS Biotechnology, Switzerland) using Superscript III and Oligo-(dT) (Invitrogen), and an Illumina-suitable amplicon was subsequently generated and sequenced by MiSeq with a 151-bp read length. All primer sequences are provided in Supplementary Table 4. Relative abundance estimation was carried out by Bowtie2-mediated alignment of the whole read pool (between 50,000 and 500,000) to unique splice variants, defined as unique sequences with a coverage of at least 0.5% of the total read count[40].

**Phylogenetic analysis.** Evolutionary relationships were inferred from a ~150 nucleotide stretch of ANP32A_X1 sequence (experimentally determined by NGS from each species) using BEAST2[41] with an HKY substitution model and a strict molecular clock. The maximum clade credibility tree was calculated with TreeAnnotator and visualized with FigTree (https://github.com/rambaut/figtree/). The exact model specifications are available from https://github.com/magnuscar/FluAdaptation.

**Mathematical modeling and predictions.** To predict the selective pressure towards avian-like or mammalian-like vPol adaptations in different host species, we extended an existing IAV dynamic model[31] (Fig. 4a). Our model follows time-development of the concentration of uninfected target cells, $U$, that become infected by isogenic IAVs expressing PB2-627E or PB2-627K at rates $\beta V_E$ or $\beta V_K$, respectively, where $V_E$ and $V_K$ describe the concentration of each virus. Cells infected with each IAV variant are denoted by $I_E$ and $I_K$. Infected cells die at rate $\delta$, and viruses are cleared at rate $c$. Cells infected with the PB2-627E virus produce new particles at rate $p_E$, while cells infected with the PB2-627K virus produce new particles at rate $p_K$. The model is formulated as a system of ordinary differential equations (ODE):

$$\frac{dU}{dt} = -\beta U(V_E + V_K)$$

$$\frac{dI_E}{dt} = \beta U V_E - \delta I_E$$

$$\frac{dI_K}{dt} = \beta U V_K - \delta I_K$$

$$\frac{dV_E}{dt} = p_E I_E - c V_E$$

$$\frac{dV_K}{dt} = p_K I_K - c V_K$$

To model passaging, we allowed the ODE model to proceed for a certain time interval and then added additional uninfected cells to the system. Each of these additions is counted as one passaging step. The virus production rates $p_E$ and $p_K$

are modeled to be dependent on the relative composition of ANP32A splice variants within cells. To this end, we estimated the virus production rates $p_{Ei}$ and $p_{Ki}$ ($i = 1,2,3$) by fitting the above model to passaging data with cells solely expressing ANP32A_X1 ($i = 1$), ANP32A_X2 ($i = 2$), or ANP32A_X3 ($i = 3$) (see Fig. 4b). We implemented the model and the fitting procedure in the statistical language R[42], with the ODE solver deSolve[43]. The scripts are available from https://github.com/magnuscar/FluAdaptation. We used parameter values for the rates that are in the same range as earlier reported values[31], namely an infection rate of $\beta = 2.7 \times 10^{-6}$ ml per virion per day, an infected cell death rate of $\delta = 4$ per day, and a virus clearance rate of $c = 3$ per day. We set the starting values for the target cells as $U_0 = 4 \times 10^5$ cells ml$^{-1}$, the infected cells $I_E = I_K = 0$ cells ml$^{-1}$, and the total viral load to $V_K + V_E = 400$ virions ml$^{-1}$. While fitting the model to the passaging data, we constrained the possible production rates of PB2-627E or PB2-627K viruses in cells solely expressing each variant ANP32A_X$i$ ($i = 1,2,3$), i.e., $p_{Ei}$ and $p_{Ki}$, to range between 0 and 200 virions per cell per day. The best estimates were obtained by minimizing the residual sum of squares between the model predictions and the data with the optim() optimizer function in R. The starting values did not have any influence on the final estimates. We obtained the 95% confidence intervals of each parameter by bootstrapping (see also https://github.com/magnuscar/FluAdaptation).

Given each species expresses $f_{X1}*100\%$ ANP32A_X1, $f_{X2}*100\%$ ANP32A_X2, and $f_{X3}*100\%$ ANP32A_X3, the combined virus production rates were calculated by:

$$p_E = f_{X1}p_{E1} + f_{X2}p_{E2} + f_{X3}p_{E3}$$

$$p_K = f_{X1}p_{K1} + f_{X2}p_{K2} + f_{X3}p_{K3}$$

The 95% confidence intervals for the combined virus production rates were determined by the weighted average of the lower bounds of the separate rates, and the upper bounds of the separate rates, respectively.

Selection of a mammalian-adaptive virus (e.g. PB2-627K) should occur when its fitness is higher than that of another variant (e.g. PB2-627E) in a particular environment (i.e. shaped by expression of different ANP32A splice variants). In a first model, we assumed the infection rate ($\beta$), infected cell death rate ($\delta$), and virus clearance rate ($c$) would be the same for the two PB2-627K/E virus variants. This is a simpler (but biologically justified) version of a more general model, in which these rates also vary depending upon the specific viral variants:

$$\frac{d}{dt}U = -\beta_E V_E U - \beta_K V_K U$$

$$\frac{d}{dt}I_E = \beta_E U V_E - \delta_E I_E$$

$$\frac{d}{dt}I_K = \beta_K U V_K - \delta_K I_K$$

$$\frac{d}{dt}V_E = p_E I_E - c_E V_E$$

$$\frac{d}{dt}V_K = p_K I_K - c_K V_K$$

The fitness of the two different viral variants can then be defined as the basic reproductive numbers:

$$R_{0,E} = \frac{\beta_E}{c_E \delta_E} \sum_{i=1}^{3} f_{Xi}p_{Ei}$$

$$R_{0,K} = \frac{\beta_K}{c_K \delta_K} \sum_{i=1}^{3} f_{Xi}p_{Ki}$$

When the basic reproductive number of the PB2-627K virus is greater than that of PB2-627E, then PB2-627K viruses will be selected. Thus, selection of PB2-627K occurs when:

$$\frac{R_{0,K}}{R_{0,E}} = \frac{\beta_K c_E \delta_E \sum_{i=1}^{3} f_{Xi}p_{Ki}}{\beta_E c_K \delta_K \sum_{i=1}^{3} f_{Xi}p_{Ei}} = \frac{\beta_K c_E \delta_E}{\beta_E c_K \delta_K} \frac{p_K}{p_E} > 1$$

If the infection rates ($\beta_K$, $\beta_E$), infected cell death rates ($\delta_K$, $\delta_E$), and virus clearance rates ($c_K$, $c_E$) are the same for both variants, selection of PB2-627K occurs when $p_K > p_E$.

To assess the risk of selecting for PB2-627K-like genotypes (e.g. mammalian adaptation) in different species, we defined a risk score, $r$, which can be interpreted as a normalized area between the constant initial PB2-627K percentage and the passaging curve. $\kappa_i$ is the percentage of PB2-627K at passage $i = 0,1,2,...,n$, where $n$ is the number of passaging steps. In typical experiments $n = 5$. The risk score is then:

$$r = \frac{1}{n\gamma} \sum_{j=1}^{n} (\kappa_j - \kappa_0)$$

where $\gamma$ is a normalizing constant with $\gamma = 100 - \kappa_0$ if $\sum_{j=1}^{n} (\kappa_j - \kappa_0) \geq 0$ or $\gamma = \kappa_0$

if $\sum_{j=1}^{n} (\kappa_j - \kappa_0) > 0$. The risk score is closer to 1 the faster PB2-627K is selected for, and closer to $-1$ the faster PB2-627E is selected for. It is 0 if no selective pressure is exerted on either variant. We summarized the predicted risk scores for the different species and estimates of $p_{Ei}$ and $p_{Ki}$ with their confidence intervals in Fig. 6a.

**Model sensitivity analyses**. To assess whether assumptions in the mathematical model impact the risk scores qualitatively, we performed three different sensitivity analyses: (i) we studied the effect of the parameter estimation procedure, and tested 360 different scenarios for the parameter ranges for $p_{Ei}$ and $p_{Ki}$ (this analysis takes into consideration the argument that these virus production rates can only range between certain upper and lower rates); (ii) we studied the effect of varying infection rate ($\beta$), the infected cell death rate ($\delta$), and the virus clearance rate ($c$), all under the assumption that these parameters are the same for both viral variants (this analysis takes into account that the values used for these rates are based on earlier estimates that show some variation[31]); (iii) we relaxed the assumption that these parameters are the same (this analysis takes into account the possibility that residue identity at PB2-627 could impact these rates). Summaries of these analyses are shown as risk score violin plots[44] in Supplementary Fig. 3.

**Bioinformatic analysis of IAV surveillance data**. All H5, H7, and H9 subtype IAV protein sequences for PB1, PB2, and PA from the indicated host species were obtained and analyzed using tools from the NIAID Influenza Research Database (IRD)[32] (http://www.fludb.org). Known, experimentally validated, mammalian-adaptive polymerase mutations[19,33] were manually selected for their appearance in magpie IAV isolates and were quantified across isolates from all other species. Abundance was estimated assuming a binomial distribution from a varying number of sequences per species. The point estimate is obtained assuming a maximum likelihood estimator for the frequency in the data, and 95% confidence intervals were estimated using the function BinomCI() from the R package DescTools, with left Wilson intervals. To test whether the abundance of mammalian-adaptive vPol substitutions in a given species is significantly higher than in gulls (the reference species for 100% ANP32A_X1 expression), we used a one-sided proportion test with a confidence level of 95% (prop.test() of R's stats package).

**Reporting summary**. Further information on research design is available in the Nature Research Reporting Summary linked to this article.

## Data availability
The authors declare that all data supporting the findings of this study are available within the paper and its supplementary information files. The source data for Figs. 1, 2, 4, and 7 and Supplementary Fig. 1 are provided as a Source Data file.

## Code availability
The customized codes and mathematical algorithms described in this paper can be freely accessed at https://github.com/magnuscar/FluAdaptation.

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

## Acknowledgements

We are grateful to Marie Pohl-Puchstein (University of Zurich, Switzerland) for human primary cell samples, Michael Huber (University of Zurich, Switzerland) for NGS advice, Adolfo García-Sastre (Icahn School of Medicine at Mount Sinai, USA) for plasmids, and Eva Moritz (University of Zurich, Switzerland) for excellent technical support. The research leading to these results received funding from the European Research Council (ERC) under EU FP7 ERC Starting Grant Agreement no. 335809 [SUMOFLU], the Swiss National Science Foundation (grant 31003A_182464), and the Novartis Foundation for Medical-Biological Research (grant 18A013) (all to BGH).

## Author contributions

Conceptualization by B.G.H. and S.St.; methodology, investigation and visualization by P.D., D.E., C.M., H.L.T., and B.G.H.; software and formal analysis by C.M., S.Sc. and O.Z.; resources by M.L. and M.B.; funding acquisition, project administration and writing by B.G.H.

## Additional information

**Competing interests:** The authors declare no competing interests.

