## [Peer Review File · Nature Communications]

Reviewers' comments:

Reviewer #1 (Remarks to the Author):

Host-specific differences in the ANP32A protein have previously been shown to be one of the factors to which influenza A viruses (IAV) must adapt when crossing from avian to mammalian species. In this study, the authors show that in birds, critical differences in the length of ANP32A are controlled through splicing and that the ratio of isoforms varies between species. Through a combination of experimental work, comparative genetics and mathematical modelling they show that differences in the ratio of ANP32A splice variants can alter the selective pressure exerted on IAVs, identify variations in splice ratios between host species and put forward a mathematical model suggesting that certain avian species could therefore maintain IAV in a 'pre-adapted' form for mammalian transmission.

The assessment of the IAV restriction imposed by different ANP32A splice variants is a useful extension of existing work in this area. The idea that differences in ANP32A splicing between species might create an elevated 'risk index' for mammalian adaptation is a fascinating one and would be extremely important if proven correct, particularly as the authors identify migratory species among those of high risk. However, three major issues make it hard to view the authors' conclusions as more than speculation.

Firstly, the model relies on a number of assumptions, notably that three key parameters are the same for both variants of the virus (beta, the likelihood of a virus initiating an infection; delta, the rate at which infected cells die; and c, the rate at which infected cells cease to become infectious). This is not entirely unreasonable but it is speculation and the model as a whole has not been experimentally tested, even at the level of minireplicon assays or viral competition experiments in tissue culture. A passing reference is made to data not shown from blackbird cells (lines 217-18) which highlights that some form of testing should have been possible.

Secondly, the authors assume that splice ratios in the particular cell lines they have screened are representative of the splice ratios in the infected tissues of organisms. This is problematic. They uncritically describe their differences observed between cell lines as being 'between species' (line 173 and subsequent). However, their own data show that a chicken cell line DF-1 (Figs 3A, C, D) is not only subject to variation depending on culturing conditions but – importantly – that it differs markedly in its levels of the X3 variant from primary chicken tissues (Fig 3B). Furthermore, the argument that expression is temperature dependent (line 183) seems hard to fit with the argument that it is stable throughout the organism, and the argument that expression is similar throughout the organism was explored in chickens, where the dominance of X1 means that variations that might

have been apparent in species such as magpies, which have a more balanced set of variants, could well have been missed. The differences seen between the avian cell lines are interesting and to speculate based on them is a very reasonable thing to do. However, on the basis of the data shown, any differences should be viewed with caution unless further testing of tissues from healthy birds can be carried out.

Thirdly, it is not clear how the selection pressures modelled would be likely to operate in nature. It is implied that maintainer species are an unappreciated risk. However, there is no indication that they could select for 'pre-adapted' IAV variants that they might have acquired from other birds or by mutation, so these would presumably be 'maintained' only at very low levels. Furthermore, the species with the highest relative risk index of harbouring pre-adapted viruses are not those (domestic poultry and waterfowl) that are typically viewed as the sources of IAV zoonoses. Were a pre-adapted virus to pass from these wild birds to domesticates, the model suggests that it would be rapidly selected to revert to its non-human-adapted form, likely neutralising the elevated risk the strain posed. Although either interpretation would be significant, at the moment I am not really clear whether we should be worried or reassured by the author's model.

Other than these points, the paper is well-written and the experiments and modelling appear to have been well-performed. However, the following minor points should also be addressed:

Minor comments:

Line 145: '(and this virus background (27))' – meaning unclear (at least to me)

Line 163: does not mention that other isoforms were also detected (for horses)

Line 229: 'some modelling parameters' – this is only a suggestion, but it might have been helpful to identify threshold parameters at which a theoretical system moves from being selective to neutral – if possible, this would provide a criterion to judge future observational data against.

Line 258: 'and validated' – where was this done?

Line 396: 'available upon request' would be preferable to deposit or make available as SI

Figures 1 and 2 make reference to a standard deviation of two datapoints. Is this correct?

Figure 3A would benefit from a phylogeny showing how the different species are related.

Figure S1A: Missing key for panels B-F.

Reviewer #2 (Remarks to the Author):

Transmission of avian influenza viruses to mammals is severely limited by strong host-range restriction barriers; however cross-species transmission is possible by acquiring the mammalian-adaptive mutation PB2-E627K. In this paper, the authors attempted to clarify whether the species' difference of host factors could be the driving force on the selection of the PB2-E627K mutation.

The authors focused on the ANP32A protein, which was reported as a host factor limiting cross-species transmission. They found that ANP32A possessed three isoforms: an avian-like isoform ANP32A_X1 with a long insert, a shorter isoform ANP32A_X2, and a mammalian-like isoform ANP32A_X3 lacking any insert. The authors then investigated the influence of these variants on virus replication by using competitive assays. Competitive passaging assays using viruses possessing PB2-627K or -627E revealed that mammalian-adaptive PB2-627K viruses were dominant in cells expressing ANP32A_X2 or ANP32A_X3, but not in ANP32A_X3-expressing cells. From these results and the expression patterns of ANP32A variants in each species, the authors established modeling methods predicting the impact of the ANP32A expression patterns on the replication of PB2-627K or -627E virus. As a result of modeling, PB2-627K viruses were predominantly selected in blackbirds, geese or swans, which were dominantly expressing ANP32A_X2 or ANP32A_X3, suggesting that these species could act as maintainers of mammalian-adaptive viruses.

Overall the experiments are well executed. However, the basic concept was previously demonstrated in cell culture and in animals (ref. 21) and molecularly supported with respect to ANP32A variants (ref. 12). Although the authors cite both papers, they did not acknowledge the fact that the concept has been reported previously. Although the identification of additional avian species with respect to potential species in which PB2-627K variants are selected based on ANP32A variant levels is new, the authors present no experimental proof. For these findings to be reported in a high-profile journal such as Nature Communications, experimental proof—by infecting candidate birds and demonstrating selection of PB2-627K variants from PB2-627E variants—is essential.

Other comments:

1. The authors did not consider ANP32B expression, which also supports the viral polymerase activity of PB2-627K.
2. Figure 1E. Expression of PB2 in the ANP32A_X1 lane is greater than that in the other lanes. This will affect the result of the pull-down assay.

3. Figure 2. The authors used human A549 cells in this study. However, since they characterized avian ANP32A for PB2-627K selection in avian hosts, the authors need to use avian cells for these experiments.
4. Figure 2. The authors need to examine whether the PB2-627K mutant dominates the PB2-627E mutant in swallow, magpie, and blackbird cells.
5. Figure 2B. The expression of endogenous ANP32A and ANP32B needs to be shown by western blotting.
6. Figure 2C–2H. The authors need to evaluate growth kinetics and competitive growth in the ANP32A-KO cells.
7. Figure 2F–2H. The authors need to test the competitive growth using a lower percentage (~1%) of PB2-627K viruses.
8. Figure 2F. PB2-627K and -627E viruses similarly propagated in ANP32A_X1 cells (see fig. 2C). Why did the PB2-627K viruses disappear after only one passage?
9. Figure 3A. Expression of ANP32A variants in normal human airway epithelial cells and A549 cells at 33 °C needs to be examined.
10. Figure 3B. Did the authors check the expression of the ANP32A variants in the cells of the chorioallantoic membrane of chicken eggs?
11. Page 6, line 119, please include the virus name.
12. Page 7, line 142, the authors state, 'we biased the system against PB2-627K'. However, we are not told in the text how this was achieved.
13. Page 8, line 160, please describe what 'certain conditions' are.

14. Page 17, line 347. What is the ratio of PB2-627E:K?

Reviewer #3 (Remarks to the Author):

The work described in the manuscript "Profiling the ANP32A Splicing Landscape Predicts Influenza A Virus Polymerase Host Adaptation" establishes ANP32A as a driver of selection and a critical predictor of PB2 evolution. The authors also establish a very clever way to evaluate the three different ANP32A splice variants and a mathematical method for prediction of what cell types that may select for the three different variants. The paper is excellent with a clear presentation of the argument which is well presented. The work is very important as it clearly presents the role of ANP32A in the selection of either the PB2 627K or E.

The edits for this paper are minor and revolve around the figures. Figure 1 features a pileup of 3 sequences that use font color to display acidic and basic residues displayed in blue and green...unfortunately, as they are also shaded in a grey box, they are indistinguishable.

Figure 4. The greyscale used to distinguish the 95% and the 20% are in some panels difficult to distinguish (Duck, Chicken). Same for "Gull" in supplemental Figure 1.

Reviewer #4 (Remarks to the Author):

The study presents a novel insight into the viral and host factors determining the host adaptation efficacy of IAV. For kinetic data analysis the authors formulate and apply mathematical model. The results are convincingly presented and provide a deeper molecular understating of the processes defining the potential of IAV for evolution.

The mathematical model-based analysis is an important tool in data interpretation. The following aspects need to be addressed.

1. Line 386: the “-“ sign needs to be put on the right hand-side of the equation for uninfected target cells (U) to describe their decline due to infection.

2. The model considers only mono-infection case. However, the coinfection with different splice variants could take place as well. Please, elaborate more on this.

3. Lines 397-398. The parameter estimates were taken from the study in which the IAV model was calibrated using in vivo data. To what extent can one use these estimates for in vitro system?

4. Lines 397-399: Units are missing. This is not acceptable for experimentally-driven modelling.

5. Line 399: “fitting procedure” - What is the statistical framework used for data fitting?

6. Lines 399-402: The parameter estimation results are not unique. What are the confidence intervals in the best-fit parameter estimates? How is the selected number of combinations (i.e., “1800”) justified?

7. Lines 420-421: The rationale behind the selection of the normalizing constant is not clear.

8. Line 422: The estimates of the $p_{E,i}$ and $p_{K,l}$ are not shown in violin plots.

Gennady Bocharov

Author rebuttal letter for NCOMMS-18-32530

We thank the 4 reviewers and editor for their constructive comments about our manuscript. Please find below all the points raised by each reviewer, with our responses indicated with a '>' symbol and in blue italics.

Reviewer #1:

Host-specific differences in the ANP32A protein have previously been shown to be one of the factors two which influenza A viruses (IAV) must adapt when crossing from avian to mammalian species. In this study, the authors show that in birds, critical differences in the length of ANP32A are controlled through splicing and that the ratio of isoforms varies between species. Through a combination of experimental work, comparative genetics and mathematical modelling they show that differences in the ratio of ANP32A splice variants can alter the selective pressure exerted on IAVs, identify variations in splice ratios between host species and put forward a mathematical model suggesting that certain avian species could therefore maintain IAV in a 'pre-adapted' form for mammalian transmission.

The assessment of the IAV restriction imposed by different ANP32A splice variants is useful extension of existing work in this area. The idea that differences in ANP32A splicing between species might create an elevated 'risk index' for mammalian adaptation is a fascinating one and would be extremely important if proven correct, particularly as the authors identify migratory species among those of high risk. However, three major issues make it hard to view the authors' conclusions as more than speculation.

Firstly, the model relies on a number of assumptions, notably that three key parameters are the same for both variants of the virus (beta, the likelihood of a virus initiating an infection; delta, the rate at which infected cells die; and c, the rate at which infected cells cease to become infectious). This is not entirely unreasonable but it is speculation and the model as a whole has not been experimentally tested, even at the level of minireplicon assays or viral competition experiments in tissue culture. A passing reference is made to data not shown from blackbird cells (lines 217-18) which highlights that some form of testing should have been possible.

*> We thank the reviewer for their overall positive assessment of our manuscript, and the importance that they place on our findings. While our model assumes that key parameters are equal between the two viral variants, we believe that this is the most legitimate approach to take for 3 reasons: (i) we use an isogenic pair of viruses that differ only by a single amino-acid substitution; (ii) the contributing parameters of our model are based on experimental data generated by passaging these two viruses; (iii) the viral PB2 protein has not previously been implicated in affecting infection rate, cell death or virion clearance. Nevertheless, we take the point of the reviewer, and to address the concern about parameter assumptions we have now performed extensive sensitivity analyses (which also addresses a suggestion from reviewer 4): we have modelled varying conditions for estimating virus production rates, varying β , δ and c , and varying β , δ and c differently for both viral variants. The results of these sensitivity analyses are now shown in new **Supplementary Figure 3** and described in lines **287-295**. Essentially, these analyses show that all potential modelling scenarios broadly support our risk*

estimates, indicating that our model assumptions are reasonable. We have also integrated our model into a publically-available user-friendly web-based platform that can be used in the future to vary all possible parameter estimates and assess the risk outcome (<https://github.com/magnuscar/FluAdaptation>). We believe that this tool will make the model most useful to the wider research community.

> The reviewer is right to suggest that we test our model, and to do this we have taken both an ‘experimental’ and ‘surveillance’ approach. In the ‘experimental’ approach, we generated 2 different cell-lines expressing defined ANP32A X1:X3 ratios (validated by NGS) and performed new competition assays between the PB2-627K/E viruses in these cells. We found that the experimentally-derived data broadly matched the predictions made by our model when taking into account the specific ANP32A X1:X3 ratios. We also tested our predictions with passaging experiments in avian cells (chicken DF-1 and LMH-1) naturally expressing defined ratios of ANP32A X1:X2:X3, and found a good correlation between empirical data and model predictions. These experimental approaches support the validity of our model in being able to make predictions where more than one ANP32A isoform is expressed, and the applicability in avian cells. These new data are described in lines 240-259 and are shown in new Figures 4C-G.

> Despite our best efforts, we could not usefully experiment on other avian cells of interest (such as swallow or magpie) due to their primary nature limiting the required cell passaging, their inability to produce suitable viral titres, and transfection/promoter specificity problems preventing mini-replicon assays. To circumvent these issues, we therefore took a ‘surveillance’ approach to test our model and looked for enrichment of known, experimentally-validated ‘mammalian-like’ adaptations in the viral polymerase complex of avian IAVs isolated from different bird species. This analysis, while unavoidably limited by the low numbers of available sequences from certain species, supports our assessment that species such as magpie have the potential to harbour IAVs that are at least partially pre-adapted to mammals. These new data and analyses are presented in Figure 7, with associated text on lines 304-331. Overall, we believe that these two independent ‘experimental’ and ‘surveillance’ validations of the model have greatly strengthened the robustness of our manuscript and our argument to expand IAV surveillance in non-traditional avian species.

Secondly, the authors assume that splice ratios in the particular cell lines they have screened are representative of the splice ratios in the infected tissues of organisms. This is problematic. They uncritically describe their differences observed between cell lines as being ‘between species’ (line 173 and subsequent). However, their own data show that a chicken cell line DF-1 (Figs 3A, C, D) is not only subject to variation depending on culturing conditions but – importantly – that it differs markedly in its levels of the X3 variant from primary chicken tissues (Fig 3B). Furthermore, the argument that expression is temperature dependent (line 183) seems hard to fit with the argument that it is stable throughout the organism, and the argument that expression is similar throughout the organism was explored in chickens, where the dominance of X1 means that variations that might have been apparent in species such as magpies, which have a more balanced set of variants, could well have been missed. The differences seen between the avian cell lines are interesting and to speculate based on them is a very reasonable thing to do. However, on the basis of the data shown, any differences should be viewed with caution unless further testing of tissues from healthy birds can be carried out.

*> The reviewer makes an excellent observation concerning differences between the cell-line data and the primary chicken tissues. Upon reviewing our protocols for the experiments concerned, we noticed that we had performed gel extraction of amplicons prior to sequencing for the primary chicken tissues, but not for the cell-lines, meaning that we likely underestimated amounts of smaller low abundance isoforms, such as X3, in these primary samples. We therefore repeated all of the sequencing using a single standardised protocol. These new data reconcile the discrepancy noted by the reviewer, and consistently show that, irrespective of primary or transformed chicken tissue, the percentage of X3 is consistent in chicken tissues at ~10%. To support this reanalysis, we have also included new independent data from another transformed chicken cell-line (LMH-1), a primary chicken cell culture (SL-29), and tissue extracted from the chorio-allantoic membrane of a 10-day old fertilized chicken egg (CAM) (asked for by reviewer 2). Analysis of all of these different chicken tissues (14 in total) shows very similar ANP32A splice variant ratios, that we believe strongly supports the notion that ratios on a gross level are largely determined by species differences. This is also supported by our original analyses of ANP32A splice variant ratios from two independent duck cell-lines (CCL-141 & SEF-R), and three independent goose cell-lines (GF-R, GN-R & CCL-169) that all show a species-specific ANP32A splice variant ratio. We have updated **Figure 3**, and the text in lines **206-210**, to make these points clearer to the reader.*

Thirdly, it is not clear how the selection pressures modelled would be likely to operate in nature. It is implied that maintainer species are an unappreciated risk. However, there is no indication that they could select for 'pre-adapted' IAV variants that they might have acquired from other birds or by mutation, so these would presumably be 'maintained' only at very low levels. Furthermore, the species with the highest relative risk index of harbouring pre-adapted viruses are not those (domestic poultry and waterfowl) that are typically viewed as the sources of IAV zoonoses. Were a pre-adapted virus to pass from these wild birds to domesticates, the model suggests that it would be rapidly selected to revert to its non-human-adapted form, likely neutralising the elevated risk the strain posed. Although either interpretation would be significant, at the moment I am not really clear whether we should be worried or reassured by the author's model.

*> We have now added additional discussion in lines **366-369** to outline a possible scenario where we believe our findings are extremely relevant. In short, we envisage that 'maintainer' and 'selector' avian species may pose an elevated risk as they have the potential to either distribute 'pre-adapted' IAVs over great distances (if migratory) or directly to mammalian species. In this regard, we therefore see these species as problematic, and perhaps overlooked, intermediate hosts between poultry/waterfowl and mammals. For example, identification of new potential intermediate hosts would have important implications for improving biocontrol measures on farms as it may be prudent to take a more active approach to limit the ability of passerine birds to contact pigs. As the reviewer points out, the species we have identified with the highest risk index are not those typically thought to play a role in zoonoses or pandemic generation, and this view in the field may be biased by current surveillance efforts.*

Other than these points, the paper is well-written and the experiments and modelling appear to have been well-performed. However, the following minor points should also be addressed:

Minor comments:

Line 145: '(and this virus background (27))' – meaning unclear (at least to me)

> We were trying to highlight, perhaps clumsily, that others have found that viral strain influences the fitness of viruses expressing PB2-627K. As this was a minor side-point we have simply deleted this phrase for clarity to the reader.

Line 163: does not mention that other isoforms were also detected (for horses)

> We have edited the sentence (lines 182-183) to mention detection of other novel isoforms.

Line 229: 'some modelling parameters' – this is only a suggestion, but it might have been helpful to identify threshold parameters at which a theoretical system moves from being selective to neutral – if possible, this would provide a criterion to judge future observational data against.

> We thank the reviewer for making this important and useful point. As suggested, we undertook an analysis to identify threshold parameters that determined the switch between selection and neutrality. This new analysis is presented as new Figure 6B with text on lines 295-302, and should be extremely useful for rapidly assessing adaptation risk in the future following determination of ANP32A splice ratios in new species. From this graph one can immediately see that, irrespective of relative X2 and X3 abundance, <30% X1 would always select for mammalian-like adaptations in the IAV polymerase (line 299). This analysis is also incorporated into the publically-available user-friendly web-based platform that we have generated (<https://github.com/magnuscar/FluAdaptation>), meaning that it is now very easy for the community to assess future observational data.

Line 258: 'and validated' – where was this done?

> Multiple validations of our model and its predictive power are now in Figure 4C-G. We have edited the text throughout to highlight the model validation experiments.

Line 396: 'available upon request' would be preferable to deposit or make available as SI

> All model codes are now available on the following GitHub page: <https://github.com/magnuscar/FluAdaptation> and the manuscript text has been updated to reflect this (including a specific 'code availability' statement).

Figures 1 and 2 make reference to a standard deviation of two datapoints. Is this correct?

> Apologies, as this was simply a typographical mistake in the figure legends – three independent experiments/datapoints contributed to these means. We have now updated all of our graphs to show the individual datapoints and have also included all of the raw data in the required 'source data file'.

Figure 3A would benefit from a phylogeny showing how the different species are related.

> As suggested, we have now generated a phylogeny of the species shown (new Figure 3E), basing this on the ANP32A_X1 sequences we determined (genomes for many of the species of interest have not yet been determined). Interestingly, the phylogeny mirrors the ANP32A splicing ratio groupings, providing another useful tool for future researchers, as one could predict that species related to swallows, magpies and blackbirds may be phenotypically similar with regards to selection of ‘mammalian-like’ polymerase adaptations.

Figure S1A: Missing key for panels B-F.

> This has now been added into the appropriate figure (now Supplementary Figure 2).

Reviewer #2:

Transmission of avian influenza viruses to mammals is severely limited by strong host-range restriction barriers; however cross-species transmission is possible by acquiring the mammalian-adaptive mutation PB2-E627K. In this paper, the authors attempted to clarify whether the species’ difference of host factors could be the driving force on the selection of the PB2-E627K mutation. The authors focused on the ANP32A protein, which was reported as a host factor limiting cross-species transmission. They found that ANP32A possessed three isoforms: an avian-like isoform ANP32A_X1 with a long insert, a shorter isoform ANP32A_X2, and a mammalian-like isoform ANP32A_X3 lacking any insert. The authors then investigated the influence of these variants on virus replication by using competitive assays. Competitive passaging assays using viruses possessing PB2-627K or -627E revealed that mammalian-adaptive PB2-627K viruses were dominant in cells expressing ANP32A_X2 or ANP32A_X3, but not in ANP32A_X3-expressing cells. From these results and the expression patterns of ANP32A variants in each species, the authors established modeling methods predicting the impact of the ANP32A expression patterns on the replication of PB2-627K or -627E virus. As a result of modeling, PB2-627K viruses were predominantly selected in blackbirds, geese or swans, which were dominantly expressing ANP32A_X2 or ANP32A_X3, suggesting that these species could act as maintainers of mammalian-adaptive viruses.

Overall the experiments are well executed. However, the basic concept was previously demonstrated in cell culture and in animals (ref. 21) and molecularly supported with respect to ANP32A variants (ref. 12). Although the authors cite both papers, they did not acknowledge the fact that the concept has been reported previously. Although the identification of additional avian species with respect to potential species in which PB2-627K variants are selected based on ANP32A variant levels is new, the authors present no experimental proof. For these findings to be reported in a high-profile journal such as Nature Communications, experimental proof—by infecting candidate birds and demonstrating selection of PB2-627K variants from PB2-627E variants—is essential.

> The reviewer is of course correct that the concept that ostriches can select for ‘mammalian-like’ adaptations in the viral polymerase was well established by Shinya et al in 2009, and a likely explanation given by the identification of ANP32A species’ differences by Long et al in 2016. We have tried to make this more explicit in the text on lines 178-180. Ostriches, like mammals, completely lack the exon required to make ANP32A_X1 or X2. We believe that the novelty of our manuscript lies in the identification of species that have the exon to make ANP32A_X1 or X2, but which nevertheless express X3 to higher levels due to differential splicing. Furthermore, the bird species we identify to have this phenotype, and which we model to impact IAV polymerase adaptation, are migratory

and/or common passerines, meaning they are more likely than ostriches to act as intermediaries to potentially pass on mammalian-adapted IAVs to mammals. We have tried to make this distinction clearer in the text on lines 360-364. A key, and novel, component to our manuscript is providing direct experimental evidence that expression of ANP32A can drive differential selection of mammalian- or avian-signature polymerases, something that has only previously been addressed by polymerase reconstitution or virus growth assays, but not by experiments simulating evolution/selection.

> To address the concern about experimental proof, we now provide experimental data to support and validate our mathematical model based on defined expression of different ratios of ANP32A splice variants (new Figure 4, including data in avian cells) – see response to reviewer 1. While we were unable to experimentally infect actual candidate birds, such as magpies or swallows, as experimental proof, we now provide data from surveillance efforts that have sequenced IAVs isolated from naturally-infected species of interest. The number of isolates is understandably low, as these species have yet to be seriously considered a host of interest, but we have found evidence that some mammalian-adaptive polymerase substitutions appear to be enriched in IAVs derived from magpies, but not other birds such as gulls or turkeys. These new analyses are presented in new Figure 7, and help to support our model and its applicability to surveillance in new hosts.

Other comments:

1. The authors did not consider ANP32B expression, which also supports the viral polymerase activity of PB2-627K.

> The reviewer makes an excellent point. While this manuscript was in revision Long et al, as well as Zhang et al, posted two independent pre-prints on biorxiv.org convincingly describing how avian ANP32B is inactive and is unable to support IAV polymerase activity:

<https://www.biorxiv.org/content/10.1101/512012v1>

<https://www.biorxiv.org/content/10.1101/529412v2>

These two independent findings indicate that the relative levels of ANP32B in avian cells, as compared to ANP32A variants, do not need to be considered in our model as ANP32B is non-functional. In addition, our passaging experiments in cells overexpressing ANP32A_X1 clearly show a selection for PB2-627E, meaning that endogenous human ANP32B, which would normally support PB2-627K selection (as shown in new Supplementary Figure 1B), has been functionally titrated-out in this experimental system. We have noted this in text lines 155-158.

2. Figure 1E. Expression of PB2 in the ANP32A_X1 lane is greater than that in the other lanes. This will affect the result of the pull-down assay.

> We thank the reviewer for highlighting this important point. Indeed, the level of PB2 seems to be consistently higher in cells that have been co-transfected with the 'stronger' binding ANP32A_X1 isoform. This is something that we have also observed in our previous manuscript (Domingues and Hale, Cell Reports, 2017). It is possible that ANP32A_X1 binding leads to stabilisation of the polymerase complex components. We have now added text to the manuscript (lines 107-109) to make this point and highlight this consistent finding to the reader.

3. Figure 2. The authors used human A549 cells in this study. However, since they characterized avian ANP32A for PB2-627K selection in avian hosts, the authors need to use avian cells for these experiments.

> As part of our model validation work, we have now performed selection experiments in chicken DF-1 and LMH-1 cells and can show that the predicted outcome is matched by empirical data in these avian models (Figures 4C-D and lines 243-249).

4. Figure 2. The authors need to examine whether the PB2-627K mutant dominates the PB2-627E mutant in swallow, magpie, and blackbird cells.

> Despite our best efforts, we could not usefully experiment on avian cells of interest (such as swallow or magpie) due to their primary nature limiting the required cell passaging, their inability to produce suitable viral titers, and transfection/promoter specificity problems preventing mini-replicon assays. To circumvent these issues we therefore took a surveillance approach and looked for enrichment of known 'mammalian-like' adaptations in the viral polymerase complexes of avian IAVs isolated from different bird species. This analysis, while unavoidably limited by the low numbers of available sequences, supports our assessment that certain avian species, such as magpies, have the potential to harbour IAVs that are at least partially pre-adapted to mammals. This new analysis is presented in new Figure 7, with associated text on lines 304-331. Overall, we believe that this validation of the model has greatly strengthened the robustness of our manuscript and provides good evidence to support our argument.

5. Figure 2B. The expression of endogenous ANP32A and ANP32B needs to be shown by western blotting.

> Western blots of endogenous ANP32A and ANP32B in parental A549s and the KO cell-line are now shown in an updated Figure 2B. As expected, while ANP32A levels are completely absent in the KO cell-line, ANP32B levels are unaffected.

6. Figure 2C–2H. The authors need to evaluate growth kinetics and competitive growth in the ANP32A-KO cells.

> We have now completed these experiments and included the new data in Supplementary Figures 1A and 1B, with related text on lines 130-132 and 155-158. The PB2-627E virus, unlike the PB2-627K virus, was severely attenuated in the parental A549-ANP32A_{KO} cell-line, confirming the inability of PB2-627E to utilize human ANP32B for replication. PB2-627K was consequently selected for in the parental A549-ANP32A_{KO} cell-line during passaging experiments, indicating that in our system, disparate phenotypes caused by overexpression of different ANP32A variants must have been a result of these constructs titrating out any effect of endogenous ANP32B.

7. Figure 2F–2H. The authors need to test the competitive growth using a lower percentage (~1%) of PB2-627K viruses.

> This was an excellent suggestion and we have now completed the requested experiments. The new data are included in Supplementary Figures 1C-E, with related text on lines 160-164. The results are essentially as predicted by the model, and are in-line with our original 20% input experiments.

8. Figure 2F. PB2-627K and -627E viruses similarly propagated in ANP32A_X1 cells (see fig. 2C). Why did the PB2-627K viruses disappear after only one passage?

> *We believe that this is due to the different sensitivities of the two assays used in original Figures 2C and 2F. In the original Figure 2C, plaque assay is used to titrate growth curve samples from 3 biologically independent replicates, and this method is not suitable for detecting replication differences of 2-3 fold. However, 2-3 fold differences in a competition assay, as performed for the original Figure 2F, would have a dramatic impact on selection. This explanation has been highlighted in the text lines 153-154.*

9. Figure 3A. Expression of ANP32A variants in normal human airway epithelial cells and A549 cells at 33 °C needs to be examined.

> *We have now done this and added the new data to Figure 3A (normal primary human epithelial cells) and Figure 3D (A549s and 33°C). As expected due to the missing exon in humans, no X1 or X2 could be found in any human tissue or at any temperature.*

10. Figure 3B. Did the authors check the expression of the ANP32A variants in the cells of the chorioallantoic membrane of chicken eggs?

> *We have now done this in response to the suggestion of the reviewer. The ANP32A variant ratios are essentially the same as we have observed in all primary chicken tissues, primary chicken cell cultures and transformed chicken cell-lines. We have updated Figure 3A and Supplementary Table 1 to include these new data. We have updated the text on lines 205-206 also.*

11. Page 6, line 119, please include the virus name.

> *This has now been added (now line 123).*

12. Page 7, line 142, the authors state, ‘we biased the system against PB2-627K’. However, we are not told in the text how this was achieved.

> *We have added the ratio used for these experiments to highlight the bias against 627K (new line 151).*

13. Page 8, line 160, please describe what ‘certain conditions’ are.

> *We have updated the text accordingly to provide more detail (new lines 177-178).*

14. Page 17, line 347. What is the ratio of PB2-627E:K?

> *These details have now been added (new lines 454-455)*

Reviewer #3:

The work described in the manuscript “Profiling the ANP32A Splicing Landscape Predicts Influenza A Virus Polymerase Host Adaptation” establishes ANP32A as a driver of selection and a critical predictor of PB2 evolution. The authors also establish a very clever way to evaluate the three different ANP32A splice variants and a mathematical method for prediction of what cell types that

may select for the three different variants. The paper is excellent with a clear presentation of the argument which is well presented. The work is very important as it clearly presents the role of ANP32A in the selection of either the PB2 627K or E.

The edits for this paper are minor and revolve around the figures. Figure 1 features a pileup of 3 sequences that use font color to display acidic and basic residues displayed in blue and green...unfortunately, as they are also shaded in a grey box, they are indistinguishable. Figure 4. The greyscale used to distinguish the 95% and the 20% are in some panels difficult to distinguish (Duck, Chicken). Same for "Gull" in supplemental Figure 1.

*> We thank the reviewer for their very positive feedback and the importance that they place on our work. We have edited the colour-schemes in **Figures 1A, 4, 5** and **Supplementary Figure 2** to assist the reader and improve clarity.*

Reviewer #4:

The study presents a novel insight into the viral and host factors determining the host adaptation efficacy of IAV. For kinetic data analysis the authors formulate and apply mathematical model. The results are convincingly presented and provide a deeper molecular understating of the processes defining the potential of IAV for evolution.

The mathematical model-based analysis is an important tool in data interpretation. The following aspects need to be addressed.

1. Line 386: the "-" sign needs to be put on the right hand-side of the equation for uninfected target cells (U) to describe their decline due to infection.

*> This typo has been corrected (new **line 502**).*

2. The model considers only mono-infection case. However, the coinfection with different splice variants could take place as well. Please, elaborate more on this.

> There might be an unintentional misunderstanding in the reviewer's comment here, and we have interpreted it to refer to co-infections between different PB2-expressing viral variants rather than 'splice variants' as written. It is true that our model does not consider cells that are co-infected with different viral variants PB2-627K and PB2-627E. We decided not to incorporate a compartment of doubly-infected cells because the production of the two different viral variants would theoretically be the same in such a cell – i.e. one PB2 protein variant may be more active depending upon the ANP32A splice variant ratio, but would nevertheless replicate the co-infecting different PB2 genomic material equally, as the selection acts at the protein level and not the genome coding level. Thus, we reason that rare co-infection events with two different viral strains would not affect overall outcome as compared to infecting two cells with the two different viral strains separately.

3. Lines 397-398. The parameter estimates were taken from the study in which the IAV model was calibrated using in vivo data. To what extent can one use these estimates for in vitro system?

> The reviewer asks a very important question on the translatability between in vivo and in vitro systems. We chose to parameterize our model with values that are supported by the literature. In

*particular, we chose this parameterization to be able to transfer the in vitro passaging results to in vivo predictions. In addition, it is not possible to determine experimentally many of the parameters for the in vitro system. However, for this reason (and also in response to a comment from reviewer 1) we have now performed additional sensitivity analyses, which allow parameter estimates, such as the infection rate (β), the cell death rate (δ) and the virus clearance rate (c), to vary for the two viral variants. The results of these sensitivity analyses are now shown in new **Supplementary Figure 3** and described in lines **560-569**. Essentially, these analyses show that all potential modelling scenarios broadly support our risk estimates, indicating that our model assumptions are reasonable.*

4. Lines 397-399: Units are missing. This is not acceptable for experimentally-driven modelling.

*> We apologise for this oversight and have updated the **text around line 510-513 accordingly**.*

5. Line 399: “fitting procedure” - What is the statistical framework used for data fitting?

*> We have now extended the explanation on the fitting procedure to provide more details (**new lines 505-510 and 518-520**)*

6. Lines 399-402: The parameter estimation results are not unique. What are the confidence intervals in the best-fit parameter estimates? How is the selected number of combinations (i.e., “1800”) justified?

*> In the revised version of the manuscript, we have altered the presentation of the analysis accordingly. We now present passaging predictions based on point estimates of the virus production rates and their 95%CI obtained by bootstrapping. We have now moved the original analysis as one of three sensitivity analyses in **Supplementary Figure 3**. Additionally, we have added a section to explain these sensitivity analyses on lines **560-569**.*

7. Lines 420-421: The rationale behind the selection of the normalizing constant is not clear.

*> The normalizing constant for the risk scores is chosen such that the risk always ranges between -1 and 1. To calculate the risk scores, we calculate the area under the curve of the passage experiment and normalize it with the maximally obtainable area. We have edited the text around lines **555-557** to clarify this for the reader.*

8. Line 422: The estimates of the $p_{E,i}$ and $p_{K,l}$ are not shown in violin plots.

*> There might be a misunderstanding. The violin plots are showing the predictions of the risk score for different values of p_{E_i} and p_{K_l} . We have reformulated the legends for **Supplementary Figure 3** to clarify this.*

REVIEWERS' COMMENTS:

Reviewer #1 (Remarks to the Author):

The authors have engaged thoughtfully and thoroughly with all of the comments I raised during the first round of review. I am satisfied with their responses to all of these points and I was reassured by the extra work done to demonstrate the robustness of the modelling. I also found the introduction of the avian phylogeny helpful, with the clarification that the species 'of concern' were all passerine birds. In my opinion the work done to address the points raised by all of the reviewers has considerably improved the manuscript and the story it now tells is clear, credible and important.

Reviewer #2 (Remarks to the Author):

The basic concept of this manuscript has already been demonstrated in vitro and in vivo. Furthermore, experimental proof, by infecting the candidate birds, is lacking in the revised manuscript. Although the authors claim that there is novelty to this manuscript in their response letter, the finding of the novel splice variants and the proposed novel model are not sufficiently innovative to grab broad scientific interest.

Reviewer #3 (Remarks to the Author):

Thank you for addressing my comments. The other features you have added to the paper only have served to strengthen the work. Nicely done and a very significant piece of work.

Reviewer #4 (Remarks to the Author):

The authors have addressed all my previous comments.

Please, clarify the following issue:

Lines 518-519: "...best estimates were obtained by minimizing the residual sum of squares between the model predictions and the data with the..."

Did the authors check that the errors in the data follow a normal distribution to apply the ordinary least-squares objective function for data fitting?

Point-by-point response to editorial requests for NCOMMS-18-32530B

We thank the reviewers and editor for their overall positive assessment of the manuscript. Please find below our point-by-point response to the comments (our responses indicated with a '>' symbol and in blue italics).

REVIEWERS' COMMENTS:

Reviewer #1 (Remarks to the Author):

The authors have engaged thoughtfully and thoroughly with all of the comments I raised during the first round of review. I am satisfied with their responses to all of these points and I was reassured by the extra work done to demonstrate the robustness of the modelling. I also found the introduction of the avian phylogeny helpful, with the clarification that the species 'of concern' were all passerine birds. In my opinion the work done to address the points raised by all of the reviewers has considerably improved the manuscript and the story it now tells is clear, credible and important.

Reviewer #2 (Remarks to the Author):

The basic concept of this manuscript has already been demonstrated *in vitro* and *in vivo*. Furthermore, experimental proof, by infecting the candidate birds, is lacking in the revised manuscript. Although the authors claim that there is novelty to this manuscript in their response letter, the finding of the novel splice variants and the proposed novel model are not sufficiently innovative to grab broad scientific interest.

Reviewer #3 (Remarks to the Author):

Thank you for addressing my comments. The other features you have added to the paper only have served to strengthen the work. Nicely done and a very significant piece of work.

Reviewer #4 (Remarks to the Author):

The authors have addressed all my previous comments.

Please, clarify the following issue:

Lines 518-519: "...best estimates were obtained by minimizing the residual sum of squares between the model predictions and the data with the..."

Did the authors check that the errors in the data follow a normal distribution to apply the ordinary least-squares objective function for data fitting?

> Unfortunately, there might be a misunderstanding. From a mathematical standpoint, errors in the data do not necessarily need to follow a normal distribution to be able to apply the ordinary least-squares objective function for data fitting. To illustrate this, we refer to Han & Wellner (The Annals of Statistics 2019, Vol 47, No 4, 2286 - 2319) who formulate the classical setting of nonparametric regression in equation (1.1) of the cited paper as:

$$Y_i = f_0(X_i) + \xi_i \text{ for } i=1, \dots, n,$$

where $f_0 \in F$, a class of possible regression functions f where $f: X \rightarrow \mathbb{R}$, X_1, \dots, X_n are i.i.d. P on (X, \mathcal{A}) , and ξ_1, \dots, ξ_n are i.i.d. "errors" independent of X_1, \dots, X_n .

*Thus, it is safe to assume that the errors in our measurements are indeed i.i.d.
Han & Wellner continue to describe that:*

the LSE [least square estimator] is known to have nice properties (e.g, rate-optimality) when:
(E) the errors $\{\xi_i\}$ are sub-Gaussian or at least subexponential

*As our analysis does not require rate-optimality, we therefore believe that it is not necessary to test
for normal distributed errors.*